# Generative Adversarial Model-Based Optimization via Source Critic Regularization

**Michael S. Yao**[*]
Department of Bioengineering
Perelman School of Medicine
University of Pennsylvania

**Yimeng Zeng**
Department of Computer
and Information Science
University of Pennsylvania

**Hamsa Bastani**
The Wharton School
University of Pennsylvania

**Jacob Gardner**
Department of Computer
and Information Science
University of Pennsylvania

**James C. Gee**
Department of Radiology
University of Pennsylvania

**Osbert Bastani**[*]
Department of Computer
and Information Science
University of Pennsylvania

## Abstract

Offline model-based optimization seeks to optimize against a learned surrogate model without querying the true oracle objective function during optimization. Such tasks are commonly encountered in protein design, robotics, and clinical medicine where evaluating the oracle function is prohibitively expensive. However, inaccurate surrogate model predictions are frequently encountered along offline optimization trajectories. To address this limitation, we propose *generative adversarial model-based optimization* using **adaptive source critic regularization (aSCR)**—a task- and optimizer- agnostic framework for constraining the optimization trajectory to regions of the design space where the surrogate function is reliable. We propose a computationally tractable algorithm to dynamically adjust the strength of this constraint, and show how leveraging aSCR with standard Bayesian optimization outperforms existing methods on a suite of offline generative design tasks. Our code is available at https://github.com/michael-s-yao/gabo.

## 1 Introduction

In many real-world tasks, we often seek to optimize the value of an objective function over some search space of inputs. Such optimization problems span across a wide variety of domains, including molecule and protein design (Guimaraes et al., 2017; Brown et al., 2019; Maus et al., 2022), patient treatment effect estimation (Kim & Bastani, 2021; Berrevoets et al., 2022; Xu & Bastani, 2023), and resource allocation in public policy (Bastani et al., 2021; Ramchandani et al., 2021). A number of algorithms have been explored for online optimization in these domains, including first-order methods, quasi-Newton methods, and Bayesian optimization (Sun et al., 2020).

However, in many situations it may prove difficult or costly to estimate the objective function for any arbitrary input configuration. Evaluating newly proposed molecules requires expensive experimental laboratory setups, and testing multiple drug doses for a single patient can potentially be dangerous. In these scenarios, the allowable budget for objective function queries is prohibitive, thereby limiting the utility of out-of-the-box online policy optimization methods.

To overcome this limitation, recent work has investigated the utility of optimization methods in the *offline* setting, where we are unable to query the objective function during the optimization

---

[*]Correspondence to: `michael.yao@pennmedicine.upenn.edu`, `obastani@seas.upenn.edu`

38th Conference on Neural Information Processing Systems (NeurIPS 2024).

process and instead only have access to a set of prior observations of inputs and associated objective values; this problem can often be referred to as offline *model-based optimization* (**MBO**) (Trabucco et al., 2021; Mashkaria et al., 2023). While one may naïvely attempt to learn a surrogate black-box model from the prior observations that approximates the true oracle objective function, such models can suffer from overestimation errors, yielding falsely promising objective estimates for inputs not contained in the offline dataset. As a result, offline optimization against the surrogate objective may yield low-scoring candidate designs according to the true oracle objective function—a key limitation of traditional policy optimization techniques in the offline setting (**Fig. 1**).

In this work, we propose a novel offline MBO algorithm that leverages source critic models to optimize a surrogate objective while simultaneously remaining in-distribution when compared against a reference offline dataset. In this setting, an optimizer is rewarded for proposing optima that are "similar" to reference data points, thereby minimizing overestimation error and allowing for more robust oracle function optimization in the offline setting. Inspired by recent work on generative adversarial networks (Goodfellow et al., 2014), we quantify design similarity by proposing a novel method that regularizes a surrogate objective model using a source critic actor, which we call *adaptive source critic regularization* (**aSCR**). We show how our algorithm can be readily leveraged with optimization methods such as Bayesian optimization (BO) and first-order methods.

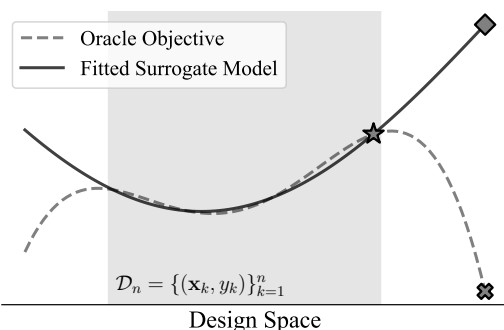

Figure 1: Naïve offline model-based optimization (MBO) (Trabucco et al., 2021), which optimizes against a learned surrogate model $f_\theta$ trained on a fixed dataset $\mathcal{D}_n = \{(\mathbf{x}_i, y_i)\}_{i=1}^n$ (shaded region) without access to the true oracle $f$, often yields candidate designs $\mathbf{x}^*$ (i.e., diamond) that score poorly using the true oracle (i.e., cross). Our method (aSCR) constrains optimization trajectories to avoid these extrapolated points, instead proposing 'in-distribution' designs (i.e., star).

**Contributions:** Our contributions are as follows: (1) We propose a novel approach for MBO that formulates the task as a constrained primal optimization problem, and we show how this framework can be used to solve for the optimal tradeoff between naïvely optimizing against the surrogate model and staying in-distribution relative to the offline dataset. (2) We introduce a computationally tractable method—which we call adaptive source critic regularization (aSCR)—to implement this framework with two popular optimization methods: Bayesian optimization and gradient ascent. (3) We show that compared to prior methods, our proposed algorithm with Bayesian optimization empirically achieves the highest rank of **3.8** (second best is 5.5) on top-1 design evaluation, and highest rank of **3.0** (second best is 4.6) on top-128 design evaluation across a variety of tasks spanning multiple scientific domains.

## 2   Related Work

Leveraging source critic model feedback for adversarial training of neural networks was popularized by works such as Goodfellow et al. (2014), where a generator and adversarial discriminator "play" a zero-sum minimax game to train a generative model. However, such discriminators often suffer from mode collapse and training instability in practice. To overcome these limitations, Arjovsky et al. (2017) introduced the Wasserstein generative adversarial net (WGAN), which instead utilizes a source critic that learns to approximate the 1-Wasserstein distance between the generated and training distributions. However, WGANs and similar networks primarily aim to generate samples that look in-distribution from a latent space prior, rather than optimize against an objective function. In our work, we adapt WGAN-inspired source critic models for Wasserstein distance estimation.

Separately in the field of optimization, Brookes et al. (2019) introduced a method for conditioning by adaptive sampling (CbAS) that learns a density model of the input space that is gradually adapted towards the optimal solution. However, such prior works have focused on solving low-dimensional, online optimization tasks (Hansen & Ostermeier, 1996; Brookes et al., 2019). More recently, Trabucco et al. (2021) introduced conservative objective models (**COM**) specifically for *offline* optimization tasks; however, their method requires directly modifying the parameters of the surrogate function

during optimization, which is not always feasible for any general task. Mashkaria et al. (2023) proposed Black-box Optimization Networks (**BONET**) to learn the dynamics of optimization trajectories using a causally masked transformer model, and Krishnamoorthy et al. (2023) introduced Denoising Diffusion Optimization Models (**DDOM**) to learn the generative process via a diffusion model. Furthermore, Yu et al. (2021) and Chen et al. (2022) describe Robust Model Adaptation (**RoMA**) and Bidirectional learning via infinite-width networks (**BDI**), respectively. RoMA regularizes the gradient of surrogate objective models by enforcing a local smoothness prior at the observed inputs, and BDI learns bidirectional mappings between low- and high- scoring candidates. Finally, Nguyen et al. (2023) introduce Experiment Pretrained Transformers (**ExPT**) to learn a general model for optimization using unsupervised methods. While these recent works and others propose promising algorithms for offline optimization tasks, they are often evaluated using expensive oracle query budgets that are often not achievable in practice—especially for potentially dangerous tasks such as patient care and other high-stakes applications.

## 3 Background

### 3.1 Offline Model-Based Optimization

In many real-world domains, we often seek to optimize an *oracle* objective function $f(\mathbf{x})$ over a space of design candidates $\mathcal{X}$ to solve for $\mathbf{x}^* = \text{argmax}_{\mathbf{x} \in \mathcal{X}} f(\mathbf{x})$. Examples of such problems include optimizing certain desirable properties of molecules in molecular design (Guimaraes et al., 2017; Brown et al., 2019; Maus et al., 2022), and estimating the optimal therapeutic intervention for patient care in clinical medicine (Kim & Bastani, 2021; Berrevoets et al., 2022; Xu & Bastani, 2023). In practice, however, the true objective function $f$ may be costly to compute or even entirely unknown, making it difficult to query in optimizing $f(\mathbf{x})$. Instead, it is often more feasible to obtain access to a reference labeled dataset of observations from nature $\mathcal{D}_n = \{(\mathbf{x}_1, y_1), \ldots, (\mathbf{x}_n, y_n)\}$ where $y_i = f(\mathbf{x}_i)$. Optimization methods may use a variety of different strategies to leverage $\mathcal{D}_n$ in the offline setting (Mashkaria et al., 2023; Krishnamoorthy et al., 2023; Chen et al., 2022); one common approach used by Trabucco et al. (2021) and others is to learn a regressor model $f_\theta$ parametrized by

$$\theta^* = \text{argmin}_\theta \ \mathbb{E}_{(\mathbf{x}_i, y_i) \sim \mathcal{D}_n} ||f_\theta(\mathbf{x}_i) - y_i||^2 \tag{1}$$

as a *surrogate model* for the true oracle objective $f(\mathbf{x})$. Rather than querying the oracle $f$ as in the online setting, we can instead solve the related optimization problem

$$\mathbf{x}^* = \text{argmax}_{\mathbf{x} \in \mathcal{X}} f_\theta(\mathbf{x}) \tag{2}$$

with the hope that optimizing $f_\theta$ will also lead to desirable oracle values of $f$ as well. Solving (2) is one instantiation of offline **model-based optimization (MBO)** for which a number of techniques have been developed, such as gradient ascent and Bayesian optimization (**BO**) (Sun et al., 2020).

Of note, it is difficult to guarantee the reliability of the model's predictions for $\mathbf{x} \notin \mathcal{D}_n$ that are almost certainly encountered in the optimization trajectory. Thus, naïvely optimizing the surrogate objective $f_\theta$ can result in "optima" that are low-scoring according to the oracle objective $f$.

### 3.2 Optimization Over Latent Spaces

In certain cases, the search space $\mathcal{X}$ for an optimization task may be discretized over a finite set of structured inputs, such as amino acids for protein sequences or atomic building blocks for molecules. However, many historical optimization algorithms do not generalize well to these settings for a number of different reasons, such as the lack of gradients with respect to the input designs to guide the optimization trajectory. Instead of directly optimizing over $\mathcal{X}$, recent work leverages deep variational autoencoders (VAEs) to first map the input space into a continuous, (often) lower dimensional latent space $\mathcal{Z}$ and then performing optimization over $\mathcal{Z}$ instead (Tripp et al., 2020; Deshwal & Doppa, 2021; Maus et al., 2022). A VAE is composed of a two components: (1) an encoder with parameters $\phi$ that learns an approximated posterior distribution $q_\phi(z|\mathbf{x})$ for $\mathbf{x} \in \mathcal{X}, z \in \mathcal{Z}$; and (2) a decoder with parameters $\varphi$ that learns the conditional likelihood distribution $p_\varphi(\mathbf{x}|z)$ (Kingma & Welling, 2013). The encoder and decoder are co-trained to maximize the evidence lower bound (ELBO)

$$\text{ELBO} = \mathbb{E}_{z \sim q_\phi} \left[ \log p_\varphi(\mathbf{x}|z) \right] - D_{\text{KL}} \left[ q_\phi(z|\mathbf{x}) \ || \ p_{\text{VAE}}(z) \right] \tag{3}$$

where $D_{\text{KL}}$ is the Kullback-Leibler (KL) divergence and $p_{\text{VAE}}(z)$ is the prior distribution. A common choice is to set $p_{\text{VAE}} = \mathcal{N}(0, I)$ (i.e., the standard normal distribution). Optimization can then be

performed over the continuous *latent space* $\mathcal{Z}$ of the VAE to propose 'latent space designs' that can be readily decoded using the decoder $\varphi$ back into the original input space.

One such optimization method over VAE latent spaces is **Bayesian optimization (BO)**, a sample-efficient framework for solving expensive black-box optimization problems (Mockus, 1982; Osborne et al., 2009; Snoek et al., 2012). While the utility of BO has primarily been explored for expensive-to-evaluate black-box functions in prior literature, recent work has shown that BO also outperforms baseline optimization methods in offline tasks involving models that are relatively inexpensive to evaluate, such as the neural network surrogates used in model-based optimization (MBO). Multiple prior works have shown that BO and related methods consistently outperform both first-order gradient-based and stochastic evolutionary methods (Eriksson et al., 2019b; Maus et al., 2022; Hvarfner et al., 2024; Eriksson & Jankowiak, 2021; Astudillo & Frazier, 2019).

### 3.3 Wasserstein Metric Between Probability Distributions

The Wasserstein distance is a distance metric between any two probability distributions, and is closely related to problems in optimal transport. We define the $p = 1$ Wasserstein distance between a reference distribution $P$ and a generated distribution $Q$ using distance metric $d(\cdot, \cdot)$ as

$$W_1(P, Q) = \inf_{\gamma \in \Gamma(P,Q)} \mathbb{E}_{(z', z) \sim \gamma} d(z', z) \tag{4}$$

where $\Gamma$ is the set of all couplings between $P$ and $Q$. For empirical distributions where $p_n$ ($q_n$) is based on $n$ observations $\{z'_j\}_{j=1}^n$ ($\{z_i\}_{i=1}^n$), (4) can be simplified to

$$W_1(p_n, q_n) = \inf_{\sigma} \frac{1}{n} \sum_{i=1}^n ||z'_{\sigma(i)} - z_i|| \tag{5}$$

where the infimum is over all permutations $\sigma$ of $n$ elements. Leveraging the Kantorovich-Rubinstein duality theorem (Kantorovich & Rubinstein, 1958), (5) can be equivalently written as

$$W_1(p_n, q_n) = \frac{1}{K} \sup_{||c||_L \leq K} \left[ \mathbb{E}_{z' \sim P}[c(z')] - \mathbb{E}_{z \sim Q}[c(z)] \right] \tag{6}$$

where $c(z)$ is a *source critic* and $||c||_L$ is the Lipschitz norm of $c(z)$. In the Wasserstein GAN (WGAN) model proposed by Arjovsky et al. (2017), a generative network and source critic are co-trained in a minimax game where the generator (critic) seeks to minimize (maximize) the Wasserstein distance $W_1$ between the training and generated distributions. Such an optimization schema enables the generator policy to learn the distribution of training samples from nature.

## 4 A Framework for Generative Adversarial Optimization

In this section we describe our proposed framework for generative adversarial model-based optimization using **adaptive source critic regularization (aSCR)**. Our method uses a $K$-Lipschitz source critic model to dynamically regularize the optimization objective to avoid extrapolation against the proxy surrogate model $f_\theta$ in offline MBO.

### 4.1 Constrained Optimization Formulation

In offline generative optimization, we aim to optimize against a surrogate objective function $f_\theta$. In order to ensure that we are achieving reliable estimates of the true, unknown oracle objective, we can add a regularization penalty to keep generated samples "similar" to those from the training dataset of $f_\theta$ according to an adversarial source critic trained to discriminate between generated and offline samples. That is, in contrast to (2), aSCR instead considers a closely related *constrained* problem

$$\begin{aligned} \text{minimize}_{z \in \mathcal{Z}} \quad & -f_\theta(z) \\ \text{subject to} \quad & \mathbb{E}_{z' \in P}[c^*(z')] - c^*(z) \leq 0 \end{aligned} \tag{7}$$

over some configuration space $\mathcal{Z} \subseteq \mathbb{R}^d$, and where we define $c^*$ as a source critic model that maximizes $\mathbb{E}_{z' \in P}[c^*(z')] - \mathbb{E}_{z \in Q}[c^*(z)]$ over all $K$-Lipschitz functions as in (6). We can think of $\mathbb{E}_{z' \in P}[c^*(z')] - c^*(z)$ as the contribution of a particular generated datum $z$ to the overall $p = 1$

Wasserstein distance between the generated candidate ($Q$) and reference ($P$) distributions of designs as in (6). In practice, we model $c^*$ as a fully connected neural net. Intuitively, the imposed constraint restricts the feasible search space to designs that score at least as in-distribution as the average sample in the offline dataset according to the source critic. Therefore, $c^*$ acts as an adversarial model to regularize the optimization policy. Of note, this additional constraint in (7) may be highly non-convex for general $c^*$, and so it is often impractical to directly apply (7) to any arbitrary MBO policy.

## 4.2 Dual Formulation

To solve this implementation problem, we instead look to reformulate (7) in its dual space by first considering the Lagrangian $\mathcal{L}$ of our constrained problem:

$$\mathcal{L}(z; \lambda) = -f_\theta(z) + \lambda \left[ \mathbb{E}_{z' \in P}[c^*(z')] - c^*(z) \right] \tag{8}$$

where $\lambda \geq 0$ is the Lagrange multiplier associated with the constraint in (7). We can equivalently think of $\lambda$ as a hyperparameter that controls the relative strength of the source critic-penalty term: $\lambda = 0$ equates to naïvely optimizing the surrogate objective, while $\lambda \gg 1$ asymptotically approaches a WGAN-like optimization policy. Minimizing $\mathcal{L}$ thus minimizes a relative sum of $-f_\theta$ and the Wasserstein distance contribution from any particular generated datum $z$ with relative weighting dictated by the hyperparameter $\lambda$. From duality, minimizing $\mathcal{L}$ over $z$ and simultaneously maximizing over $\lambda \in \mathbb{R}_+$ is equivalent to the original constrained problem in (7).

The challenge now is in determining this optimal value of $\lambda$: if $\lambda$ is too small, then the objective estimates may be unreliable; if $\lambda$ is too large, then the optimization trajectory may be unable to adequately explore the input space. Prior work by Trabucco et al. (2021) has previously explored the idea of formulating offline optimization problems as a similarly regularized Lagrangian (albeit with a separate regularization constraint), although their method tunes a static hyperparameter by hand. In contrast, aSCR treats $\lambda$ as a dynamic parameter that adapts to the optimization trajectory in real time.

## 4.3 Computing the Lagrange Multiplier $\lambda$

Continuing with our dual formulation of (7), the Lagrange dual function $g(\lambda)$ is defined as $g(\lambda) = \inf_{z \in \mathbb{R}^n} \mathcal{L}(z; \lambda)$. The $z = \hat{z}$ that minimizes the Lagrangian in the definition of $g$ is evidently a function of $\lambda$. To show this, we use the first-order condition that $\nabla_z \mathcal{L} = 0$ at $z = \hat{z}$. Per (8), we have

$$\nabla_z \mathcal{L}(\hat{z}; \lambda) = -\nabla_z f_\theta(\hat{z}) - \lambda \nabla_z c^*(\hat{z}) = 0 \tag{9}$$

In general, solving (9) for $\hat{z}$ is computationally intractable—especially in high-dimensional problems. Instead, we can approximate $\hat{z}$ by relaxing the condition in (9) according to

$$\hat{z}(\lambda) = \operatorname*{argmin}_{z \in \mathbb{R}^n} \frac{1}{2} \left\| -\nabla_z f_\theta(z) - \lambda \nabla_z c^*(z) \right\|^2 \tag{10}$$

Our key insight is that although minimizing the loss term in (10) is not practical when the feasible set is naïvely uniform over $\mathbb{R}^n$, we can instead choose to focus our attention on latent space coordinates with high associated probability according to the VAE prior distribution $p_{\text{VAE}}(z)$. This is because in optimization problems acting over the latent space of any variational autoencoder, the majority of the encoded information content is embedded according to $p_{\text{VAE}}(z)$ due to the Kullback-Leibler (KL) divergence contribution to VAE training. Put simply, the encoder distribution $q_\phi(z|\mathbf{x})$ is trained so that $D_{\text{KL}}[q_\phi(z|\mathbf{x})||p_{\text{VAE}}(z))]$ is optimized as a regularization term in (3). We argue that it is thus sufficient enough to approximate $\hat{z}(\lambda)$ using a Monte Carlo sampling schema with random samples $\mathcal{Z}_N = (z_1, z_2, \ldots, z_N) \sim p_{\text{VAE}}(z)$:

$$\hat{z}(\lambda) \approx \operatorname*{argmin}_{\mathcal{Z}_N \sim p_{\text{VAE}}(z)} \frac{1}{2} \left\| -\nabla_z f_\theta(z) - \lambda \nabla_z c^*(z) \right\|^2 \tag{11}$$

We can now concretely write an approximation of the Lagrange dual problem of (7):

$$
\begin{aligned}
\text{maximize} \quad & g(\lambda) = -f_\theta(\hat{z}) + \lambda \left[ \mathbb{E}_{z' \in P}[c^*(z')] - c^*(\hat{z}) \right] \\
\text{subject to} \quad & \lambda \geq 0
\end{aligned}
\tag{12}
$$

where $\hat{z}$ is as in (11). Defining the surrogate variable $\alpha$ such that $\lambda = \frac{\alpha}{1-\alpha}$, we can rewrite (12) as

$$
\begin{aligned}
\text{maximize} \quad & -(1-\alpha)f_\theta(\hat{z}) + \alpha \left[ \mathbb{E}_{z' \in P}[c^*(z')] - c^*(\hat{z}) \right] \\
\text{subject to} \quad & 0 \leq \alpha < 1
\end{aligned}
\tag{13}
$$

In practice, we discretize the search space for $\alpha$ to 200 evenly spaced points between 0 and 1 inclusive. From weak duality, finding the optimal solution to (12) provides a lower bound on the optimal solution to the primal problem in (7). **Algorithm 1** can now be used to choose the optimal $\alpha$ (and hence $\lambda$) adaptively during offline optimization: we refer to our method as **Adaptive SCR (aSCR)**.

### 4.4 Overall Algorithm

Using Adaptive SCR, we now have a proposed method for dynamically computing $\alpha$ (and hence the Lagrange multiplier $\lambda$) of the constrained optimization problem in (7). Importantly, aSCR can be integrated with any standard function optimization method by optimizing the Lagrangian objective in (8) over the candidate design space as opposed to the original unconstrained objective $f_\theta$. We refer to this algorithm as *Generative Adversarial Model-Based Optimization* (GAMBO). To evaluate aSCR empirically, we instantiate two flavors of GAMBO: (1) **G**enerative **A**dversarial **B**ayesian **O**ptimization (**GABO**, **Algorithm 2**); and (2) **G**enerative **A**dversarial **G**radient **A**scent (**GAGA**).[2]

We implement GABO using a quasi-expected improvement (qEI) acquisition function, iterative sampling budget of $T = 32$, sampling batch size of $b = 64$, and GAGA using a step size of $\eta = 0.05$, $T = 128$, and $b = 16$. Of note, the optimization objective using aSCR is time-varying and causally linked to past observations made during the optimization process via intermittent training of the source critic $c$. Prior works from Nyikosa et al. (2018) and Aglietti et al. (2022) have examined optimization against dynamic objective functions, although have either entirely disregarded causal relationships between variables or only examined causality between inputs as opposed to inputs and the objective. We leave such methods for future work given that aSCR works well in practice.

## 5 Experimental Evaluation

### 5.1 Datasets and Tasks

To evaluate our proposed algorithm, we focus on a set of eight tasks spanning multiple domains with publicly available datasets in the field of offline model-based optimization. (1) The **Branin** function is a well-known synthetic benchmark function where the task is to maximize the two-dimensional Branin function $f_{br} : [-5, 10] \times [0, 15] \to \mathbb{R}$. (2) The **LogP** task is a well-studied optimization problem (Zhou et al., 2019; Chen et al., 2021; Flam-Shepherd et al., 2022) where we search over candidate molecules to maximize the penalized water-octanol partition coefficient (logP) score, which is an approximate measure of a molecule's hydrophobicity (Ertl & Schuffenhauer, 2009) that also rewards structures that can be synthesized easily and feature minimal ring structures. We use the publicly available Guacamol benchmarking dataset from Brown et al. (2019) to implement this task.

Tasks (3) - (7) are derived from Design-Bench, a publicly available set of MBO benchmarking tasks (Trabucco et al., 2022): (3) **TF-Bind-8** aims to maximize the transcription factor binding efficiency of an 8-base-pair DNA sequence (Barrera et al., 2016); (4) **GFP** the green fluorescence of a 237-amino-acid protein sequence (Brookes et al., 2019; Rao et al., 2019); (5) **UTR** the gene expression from a 50-base-pair 5'UTR DNA sequence (Sample et al., 2019; Angermueller et al., 2020); (6) **ChEMBL** the mean corpuscular hemoglobin concentration (MCHC) biological response of a molecule using an offline dataset collected from the ChEMBL assay `CHEMBL3885882` (Gaulton et al., 2012); and (7) **D'Kitty** the morphological structure of the D'Kitty robot (Ahn et al., 2020).

Finally, (8) the **Warfarin** task uses the dataset of patients on warfarin medication from Consortium (2009) to estimate the optimal dose of warfarin given clinical and pharmacogenetic patient data. Of note, in contrast to tasks (1) - (7) and other traditional MBO tasks in prior literature (Trabucco et al., 2022), the Warfarin task is novel in that only a subset of the input design dimensions may be optimized over (i.e., warfarin dose) while the others remain fixed as conditioning variables (i.e., patient covariates). Such a task can therefore be thought of as *conditional* model-based optimization.

### 5.2 Policy Optimization and Evaluation

For all experiments, the surrogate objective model $f_\theta$ is a fully connected net with two hidden layers of size 2048 and LeakyReLU activations. $f_\theta$ takes as input a VAE-encoded latent space datum and

---

[2]We detail the explicit algorithmic formulation for GAGA in **Supplementary Algorithm 3**.

returns the predicted objective function value as output. The VAE encoder and decoder backbone architectures vary by MBO task and are detailed in **Supplementary Table A1**. Following Gómez-Bombarelli et al. (2018) and Maus et al. (2022), we co-train the VAE and surrogate objective models together using an Adam optimizer with a learning rate of $3 \times 10^{-4}$ for all tasks. For the optimization tasks over continuous design spaces (i.e., Branin, Warfarin, and D'Kitty), we fix the VAE encoder and decoders as the identity functions, such that the latent and input spaces are equivalent.

The source critic agent $c$ in (7) is implemented as a fully connected net with two hidden layers with sizes equal to four (one) times the number of input dimensions for the first (second) layer. To constrain the Lipschitz norm of $c$ as in (6), we clamp the weights of the model between [-0.01, 0.01] after each optimization step as done by Arjovsky et al. (2017). The model is trained using gradient descent with a learning rate of 0.001 to maximize the Wasserstein distance between the dataset and generated candidates in the VAE latent space.

During optimization, both GABO and GAGA alternate between sampling new designs and training the source critic actor $c(z)$ until there is no improvement to the Wasserstein distance $W_1$ according to $c$ after 100 consecutive weight updates. We find that training $c$ every $n_{\text{generator}} = 4$ sampling steps is a good choice across all tasks assessed, similar to prior work Arjovsky et al. (2017).

All MBO methods were evaluated using a fixed surrogate query budget of 2048. We focus on two evaluation metrics: 100th percentile (1) top $k = 1$; and (2) top $k = 128$ oracle score. The top $k = 128$ evaluation metric is commonly reported in prior offline MBO literature (Mashkaria et al., 2023; Trabucco et al., 2021; Yu et al., 2021); the top $k = 1$ metric better accounts for the limited oracle query budget of the real-world tasks in which offline MBO would be of use. In both settings, an optimizer selects the top $k$ design that minimize the Lagrangian function value in (8) from the 2048 assessed designs to evaluate using the true oracle function, and the maximum score of those $k$ designs is reported across 10 random seeds.

We evaluate both GABO and GAGA against a number of pre-existing baseline algorithms on one internal cluster with 8 NVIDIA RTX A6000 GPUs. We include vanilla Bayesian Optimization (**BO**-qEI) and gradient ascent (**Grad.**) in our evaluation to assess the utility of our proposed aSCR algorithm. Furthermore, we evaluate limited-memory BFGS (**L-BFGS**) Liu & Nocedal (1989), **CMA-ES** Hansen & Ostermeier (1996), and

---

**Algorithm 1** Adaptive Source Critic Regularization (SCR)

**Input:** differentiable surrogate objective $f_\theta : \mathbb{R}^d \to \mathbb{R}$, differentiable source critic $c : \mathbb{R}^d \to \mathbb{R}$, reference dataset $\mathcal{D}_n = \{z'_j\}_{j=1}^n$, $\alpha$ step size $\Delta\alpha$, search budget $\mathcal{B}$, norm threshold $\tau$
Sample candidates $\mathcal{Z}_\mathcal{B} \leftarrow \{z_i\}_{i=1}^\mathcal{B} \sim \mathcal{N}(0, I_d)$
Initialize $\alpha^* \leftarrow$ None and $g^* \leftarrow -\infty$
**for** $\alpha$ **in** range(start $= 0$, end $= 1$, stepsize $= \Delta\alpha$) **do**
    $z^* \leftarrow \text{argmin}_{z_i \in \mathcal{Z}_\mathcal{B}} ||(1-\alpha)\nabla f_\theta(z_i) + \alpha\nabla c(z_i)||_2$
    **if** $||(1-\alpha)\nabla f_\theta(z^*) + \alpha\nabla c(z^*)||_2 > \tau$ **then**
        **continue**        // Discard $\alpha$ if best norm exceeds $\tau$
    **end if**
    $g \leftarrow -(1-\alpha)f_\theta(z^*) + \alpha\left[\mathbb{E}_{\mathcal{D}_n}[c(z'_j)] - c(z^*)\right]$
    **if** $g > g^*$ **then**
        $\alpha^* \leftarrow \alpha$ and $g^* \leftarrow g$        // Implements (13)
    **end if**
**end for**
**return** $\alpha^*$

---

**Algorithm 2** Generative Adversarial BayesOpt (GABO)

**Input:** surrogate objective $f_\theta : \mathbb{R}^d \to \mathbb{R}$, offline dataset $\mathcal{D}_n = \{z'_j\}_{j=1}^n$, acquisition function $a$, iterative sampling budget $T$, sampling batch size $b$, number of generator steps per source critic training $n_{\text{generator}}$, oracle query budget $k$
**AdaptiveSCR Input:** $\alpha$ step size $\Delta\alpha$, search budget $\mathcal{B}$, norm threshold $\tau$
**Define:** Differentiable source critic $c : \mathbb{R}^d \to \mathbb{R}$
**Define:** Lagrangian $\mathcal{L}(z; \alpha) : \mathbb{R}^d \times \mathbb{R} \to \mathbb{R}$        // Eq. (8)
    $\mathcal{L}(z; \alpha) = -f_\theta(z) + \frac{\alpha}{1-\alpha}[\mathbb{E}_{z' \sim \mathcal{D}_n}[c(z')] - c(z)]$
Sample candidates $\mathcal{Z}^1 \leftarrow \{z_i^1\}_{i=1}^b \sim$ SobolSequence
// Train the source critic per Eq. (6) to optimality:
$c \leftarrow \text{argmax}_{||c||_L \leq K} W_1(\mathcal{D}_n, \mathcal{Z}^1)$
    $= \text{argmax}_{||c||_L \leq K} [\mathbb{E}_{z' \sim \mathcal{D}_n}[c(z')] - \mathbb{E}_{z \sim \mathcal{Z}^1}[c(z)]]$
$\alpha \leftarrow$ **AdaptiveSCR**$(f_\theta, c, \mathcal{D}_n, \Delta\alpha, \mathcal{B}, \tau)$        // Alg. (1)
Evaluate candidates $\mathcal{Y}^1 \leftarrow \{y_i^1\}_{i=1}^b = \{-\mathcal{L}(z_i^1; \alpha)\}_{i=1}^b$
Place Gaussian Process (GP) prior on $f_\theta$
**for** $t$ **in** $2, 3, \dots, T$ **do**
    Update posterior on $f_\theta$ with $\mathcal{D}_{t-1} = \{(\mathcal{Z}^m, \mathcal{Y}^m)\}_{m=1}^{t-1}$
    Compute acquisition function $a$ using fitted posterior
    Sample candidates $\mathcal{Z}^t \leftarrow \{z_i^t\}_{i=1}^b$ according to $a$
    $\alpha \leftarrow$ **AdaptiveSCR**$(f_\theta, c, \mathcal{D}_n, \Delta\alpha, \mathcal{B}, \tau)$
    Evaluate samples $\mathcal{Y}^t \leftarrow \{y_i^t\}_{i=1}^b = \{-\mathcal{L}(z_i^t; \alpha)\}_{i=1}^b$
    **if** $t$ mod $n_{\text{generator}}$ equals 0 **then**
        // Train the source critic per Eq. (6) to optimality:
        $c \leftarrow \text{argmax}_{||c||_L \leq K} W_1(\mathcal{D}_n, \mathcal{Z}^t)$
            $= \text{argmax}_{||c||_L \leq K} [\mathbb{E}_{z' \sim \mathcal{D}_n}[c(z')] - \mathbb{E}_{z \sim \mathcal{Z}^t}[c(z)]]$
    **end if**
**end for**
**return** the top $k$ samples from the $T \times b$ observations
    $\mathcal{D}_T = \{\{(z_i^m, y_i^m)\}_{i=1}^b\}_{m=1}^T$ according to $y_i^m$

---

simulated annealing (**Anneal**) Kirkpatrick et al. (1983). We also compare our method against the more recently introduced algorithms **TuRBO**-qEI (Eriksson et al., 2019a), **COM** (Trabucco et al.,

2021), **RoMA** (Yu et al., 2021), **BDI** (Chen et al., 2022), **DDOM** (Krishnamoorthy et al., 2023), **BONET** (Mashkaria et al., 2023), **ExPT** (Nguyen et al., 2023), **ROMO** (Chen et al., 2023), and **BootGen** (Kim et al., 2023). Because BootGen is proposed by Kim et al. (2023) as an optimization method specifically for biological sequence design, we only assess this baseline method on the five relevant tasks in our evaluation suite.

**Conditional MBO Tasks.** To our knowledge, prior work in conditional model-based optimization is limited, and so previously reported algorithms are not equipped to solve such tasks out-of-the-box. Chen et al. (2023) explore such tasks in their work, but primarily focus on conditional tasks that are built by arbitrarily fixing certain design dimensions from unconstrained problems, which are not representative of true conditional optimization problems in the real world. In our work, we introduce the Warfarin task to assess methods on their ability to design an optimal therapeutic drug regiment *conditioned* on a fixed patient state and lab values. To assess existing methods on this task, we implement conditional proxies of all baselines employing a first-order optimization schema via *partial* gradient ascent to only update the warfarin dose dimension while leaving the patient attribute conditional dimensions unchanged. Conditional BO-based methods are implemented by fitting separate Gaussian processes for each patient. In conditional DDOM, we exchange the algorithm's diffusion model-based backbone with a *conditional* score-based diffusion model (Gu et al., 2023).

Of note, the BONET algorithm (Mashkaria et al., 2023) requires multiple observations for any given patient to construct synthetic optimization trajectories. However, the key challenge in conditional MBO is that each condition (i.e., patient) has *no* past observations (i.e., warfarin doses), and instead relies on learning from offline datasets constructed from different permutations of condition values As a result, the BONET algorithm is unable to be evaluated on conditional MBO tasks.

### 5.3   Main Results

Scoring of one-shot optimization candidates is shown in **Table 1**. Across all eight assessed tasks spanning a wide range of scientific domains, GABO with our aSCR algorithm achieved the best average rank of **3.8** when compared to other existing methods (next best is 5.5). Furthermore, GABO was able to propose top $k = 1$ candidate designs that outperform the best design in the pre-existing offline dataset for 6 of the 8 tasks–greater than any of the other methods assessed. If a larger oracle evaluation budget is available (i.e., $k = 128$), GABO with aSCR performs even better, achieving the best average rank of **3.0** (next best is 4.6). GABO is also the best algorithm on 3 of the 8 tasks and second best on 2 tasks according to this evaluation metric. Altogether, our results suggest that GABO is a promising method for proposing optimal design candidates in offline MBO.

Importantly, our aSCR algorithm improves upon both the naïve BO-qEI and Grad. Ascent parent optimizers assessed. GABO outperforms both baseline BO-based optimization methods in our evaluation suite: BO (TuRBO) only achieves a rank of 8.8 (9.0) on the top $k = 1$ evaluation metric and a rank of 6.6 (7.4) on the top $k = 128$ metric. Similarly, GAGA scores an average rank of 7.4 (7.6) on the top $k = 1$ ($k = 128$) evaluation metric; by leveraging aSCR, GAGA outperforms its base parent optimizer (Grad. Ascent), which only achieves an average rank of 9.0 and 11.0 on the same two evaluation metrics, respectively. Our results show that using aSCR to adaptively penalize the objective of two popular optimization methods can improve their offline performance.

**Qualitative Evaluation: Penalized LogP Task.** We evaluate GABO against naïve BO-qEI for the **LogP** task by inspecting the three-dimensional chemical structures of the top-scoring candidate molecules. As a general principle, molecules that are associated with high Penalized LogP scores are hydrophobic with minimal ring structures and therefore often feature long hydrocarbon backbones (Ertl & Schuffenhauer, 2009). In **Figure 2**, we see that BO-qEI using the unconstrained surrogate objective generates a candidate molecule of hydrogen and carbon atoms. However, the proposed candidate includes two rings in its structure, resulting in a suboptimal oracle Penalized LogP score.

We hypothesize that this may be due to a lack of ring-containing example molecules in the offline dataset, as only 6.7% (2.7%) of observed molecules contain at least one (two) carbon ring(s). As a result, the surrogate objective model estimator returns more inaccurate Penalized LogP estimates for input ring-containing structures (surrogate model root mean squared error (RMSE) = 25.5 for offline dataset molecules with at least 2 rings; RMSE = 16.5 for those with at least 1 ring; and RMSE = 4.6 for those with at least 0 rings), leading to sub-par BO-qEI optimization performance as the unconstrained algorithm extrapolates against the surrogate to find "optimal" molecules that

Table 1: **Constrained Budget** ($k = 1$) **Oracle Evaluation**     Each method proposes a single design that is evaluated using the oracle function to report the final score (higher is better) across 10 random seeds reported as mean ± standard deviation. $\mathcal{D}$ (best) reports the top oracle value in the task dataset. Each of the MBO methods are ranked by their mean one-shot oracle score, and the average rank (lower is better) across all eight tasks is reported in the final table column. **Bold** (Underlined) entries indicate the best (second best) entry in the column. *Denotes the life sciences-related discrete MBO tasks from Design-Bench (Trabucco et al., 2022).

| Method | Branin | LogP | TF-Bind-8* | GFP* | UTR* | ChEMBL* | D'Kitty | Warfarin | Rank |
|---|---|---|---|---|---|---|---|---|---|
| $\mathcal{D}$ (best) | -13.0 | 11.3 | 0.439 | 3.53 | 7.12 | 0.61 | 0.88 | -0.19 ± 1.96 | — |
| Grad. | -245.1 ± 81.3 | -5.37 ± 1.44 | 0.429 ± 0.023 | 3.18 ± 0.88 | 6.82 ± 0.21 | -1.95 ± 0.00 | 0.57 ± 0.19 | 0.86 ± 1.09 | 9.0 |
| L-BFGS | -29.6 ± 0.0 | 3.82 ± 32.6 | 0.527 ± 0.140 | 3.51 ± 0.70 | 6.48 ± 1.20 | -1.95 ± 0.00 | 0.31 ± 0.00 | 0.73 ± 1.83 | 8.5 |
| CMA-ES | -8.6 ± 3.6 | 5.04 ± 6.83 | 0.438 ± 0.131 | 1.43 ± 0.00 | 6.39 ± 0.11 | -1.95 ± 0.00 | 0.31 ± 0.00 | -25.0 ± 150 | 10.6 |
| Anneal | -9.6 ± 1.5 | 8.76 ± 0.15 | 0.807 ± 0.094 | 3.64 ± 0.03 | 5.01 ± 0.31 | -1.95 ± 0.00 | 0.55 ± 0.18 | **0.91 ± 0.08** | 6.8 |
| BO | -11.0 ± 7.8 | -52.5 ± 88.8 | 0.586 ± 0.193 | 1.43 ± 0.00 | 5.65 ± 1.30 | 0.59 ± 0.10 | 0.61 ± 0.15 | 0.16 ± 1.67 | 8.8 |
| TuRBO | -21.0 ± 5.1 | -45.1 ± 93.8 | 0.564 ± 0.194 | 1.43 ± 0.00 | 6.53 ± 1.19 | **0.65 ± 0.00** | 0.44 ± 0.18 | 0.05 ± 0.11 | 9.0 |
| BONET | -26.1 ± 0.9 | 10.8 ± 0.33 | 0.282 ± 0.000 | **3.74 ± 0.00** | **9.12 ± 0.07** | 0.55 ± 0.13 | 0.78 ± 0.00 | — | 5.7 |
| DDOM | -6677 ± 6360 | -4.23 ± 1.28 | 0.460 ± 0.030 | 1.43 ± 0.00 | 5.56 ± 0.02 | 0.54 ± 0.15 | 0.51 ± 0.20 | -0.32 ± 0.40 | 11.1 |
| COM | -3099 ± 32.6 | **30.8 ± 19.5** | 0.439 ± 0.000 | 3.62 ± 0.00 | 6.65 ± 0.43 | 0.63 ± 0.01 | **0.90 ± 0.02** | 0.72 ± 0.97 | 5.5 |
| RoMA | -32.7 ± 18.4 | 6.37 ± 1.39 | 0.433 ± 0.040 | 3.37 ± 0.27 | 6.66 ± 0.98 | 0.50 ± 0.14 | 0.30 ± 0.27 | -0.70 ± 0.02 | 9.4 |
| BDI | -1050 ± 0.0 | -0.20 ± 0.00 | 0.311 ± 0.000 | 3.26 ± 0.82 | 5.61 ± 0.00 | 0.48 ± 0.00 | 0.67 ± 0.00 | -24.8 ± 233 | 10.8 |
| ExPT | -57.2 ± 38.6 | -15.9 ± 24.1 | 0.571 ± 0.076 | 1.43 ± 0.00 | 6.77 ± 1.38 | 0.56 ± 0.06 | 0.66 ± 0.20 | -34.6 ± 61.4 | 9.1 |
| BootGen | — | -13.0 ± 15.1 | **0.942 ± 0.022** | 3.10 ± 0.73 | 8.30 ± 0.93 | 0.59 ± 0.07 | — | — | 6.2 |
| ROMO | -2614 ± 739.9 | -20.5 ± 19.2 | 0.382 ± 0.203 | 3.55 ± 0.13 | 5.73 ± 1.42 | **0.65 ± 0.00** | 0.64 ± 0.27 | -0.71 ± 2.10 | 9.6 |
| **GAGA** | -2.9 ± 2.2 | -68.6 ± 109.8 | 0.571 ± 0.120 | **3.74 ± 0.00** | 5.89 ± 1.42 | -1.95 ± 0.00 | 0.89 ± 0.00 | 0.01 ± 0.14 | 7.4 |
| **GABO** | **-2.6 ± 1.1** | 21.3 ± 33.2 | 0.570 ± 0.131 | 3.60 ± 0.40 | 7.51 ± 0.39 | 0.60 ± 0.07 | 0.71 ± 0.01 | 0.60 ± 1.80 | **3.8** |

Table 2: **Relaxed Budget** ($k = 128$) **Oracle Evaluation**     Each method now proposes 128 designs that are evaluated using the oracle function, and maximum score out of these 128 designs is reported below (averaged across 10 random seeds and reported as mean ± standard deviation). $\mathcal{D}$ (best) reports the top oracle value in the task dataset. Each of the MBO methods are ranked by their mean $k = 128$-shot oracle score, and the average rank (lower is better) across all eight tasks is reported in the final table column. **Bold** (Underlined) entries indicate the best (second best) entry in the column. *Denotes the life sciences-related discrete MBO tasks from Design-Bench (Trabucco et al., 2022).

| Method | Branin | LogP | TF-Bind-8* | GFP* | UTR* | ChEMBL* | D'Kitty | Warfarin | Rank |
|---|---|---|---|---|---|---|---|---|---|
| $\mathcal{D}$ (best) | -13.0 | 11.3 | 0.439 | 3.53 | 7.12 | 0.61 | 0.88 | -0.19 ± 1.96 | — |
| Grad. | -115.3 ± 20.8 | -5.14 ± 1.70 | 0.977 ± 0.025 | 3.49 ± 0.69 | 7.38 ± 0.15 | -1.95 ± 0.00 | 0.87 ± 0.02 | 0.86 ± 1.08 | 11.0 |
| L-BFGS | -4.0 ± 0.0 | 42.8 ± 9.44 | 0.633 ± 0.140 | **3.74 ± 0.00** | 7.51 ± 0.39 | -1.95 ± 0.00 | 0.31 ± 0.00 | 0.75 ± 1.67 | 10.1 |
| CMA-ES | -4.3 ± 1.7 | 47.6 ± 5.46 | 0.810 ± 0.235 | **3.74 ± 0.00** | 7.40 ± 0.32 | -1.95 ± 0.00 | 0.74 ± 0.00 | -8.62 ± 63.8 | 9.8 |
| Anneal | -7.4 ± 2.8 | 11.3 ± 0.00 | 0.890 ± 0.035 | 3.72 ± 0.00 | 7.96 ± 0.22 | -1.95 ± 0.00 | 0.88 ± 0.00 | 0.97 ± 0.08 | 9.3 |
| BO | **-0.4 ± 0.0** | **135.3 ± 16.0** | 0.942 ± 0.025 | 2.26 ± 1.03 | 8.26 ± 0.09 | 0.67 ± 0.00 | 0.72 ± 0.00 | 0.93 ± 0.11 | 6.6 |
| TuRBO | -0.7 ± 0.4 | 59.7 ± 51.3 | 0.895 ± 0.049 | 1.89 ± 0.92 | 8.26 ± 0.11 | 0.67 ± 0.01 | 0.72 ± 0.00 | 0.99 ± 0.01 | 7.4 |
| BONET | -26.0 ± 0.9 | 11.7 ± 0.38 | 0.951 ± 0.035 | **3.74 ± 0.00** | 9.13 ± 0.08 | 0.67 ± 0.01 | 0.95 ± 0.01 | — | 5.6 |
| DDOM | -18.4 ± 29.8 | -2.16 ± 0.60 | 0.936 ± 0.051 | 1.44 ± 0.00 | 8.30 ± 0.33 | 0.66 ± 0.01 | 0.89 ± 0.01 | **1.00 ± 0.00** | 8.4 |
| COM | -1981 ± 224.5 | 42.0 ± 16.9 | 0.902 ± 0.056 | 3.62 ± 0.00 | 8.18 ± 0.00 | 0.64 ± 0.01 | 0.95 ± 0.02 | 0.77 ± 0.86 | 8.5 |
| RoMA | -4.8 ± 3.0 | 10.8 ± 0.78 | 0.760 ± 0.113 | **3.74 ± 0.00** | 8.12 ± 0.09 | 0.69 ± 0.03 | **1.02 ± 0.04** | 0.67 ± 0.05 | 7.8 |
| BDI | -65.0 ± 51.3 | 1.52 ± 5.79 | 0.735 ± 0.086 | 3.61 ± 0.05 | 6.31 ± 0.00 | 0.50 ± 0.12 | 0.94 ± 0.01 | -5.07 ± 21.0 | 11.8 |
| ExPT | -1.7 ± 1.0 | -6.48 ± 4.58 | 0.927 ± 0.095 | **3.74 ± 0.00** | 8.13 ± 0.09 | 0.68 ± 0.04 | 0.97 ± 0.01 | 0.96 ± 0.05 | 6.5 |
| BootGen | — | 8.10 ± 3.31 | **0.979 ± 0.002** | **3.74 ± 0.00** | **10.5 ± 0.95** | 0.68 ± 0.00 | — | — | 4.6 |
| ROMO | -2367 ± 787.5 | -6.05 ± 14.5 | 0.572 ± 0.202 | 3.67 ± 0.03 | 6.94 ± 1.07 | 0.65 ± 0.00 | 0.90 ± 0.02 | 0.76 ± 1.91 | 12.1 |
| **GAGA** | -1.0 ± 0.2 | 14.1 ± 25.0 | 0.722 ± 0.091 | **3.74 ± 0.00** | 7.98 ± 0.36 | -1.95 ± 0.00 | 0.90 ± 0.01 | 0.95 ± 0.07 | 7.6 |
| **GABO** | -0.5 ± 0.1 | 122.1 ± 20.6 | 0.954 ± 0.025 | **3.74 ± 0.00** | 8.36 ± 0.08 | **0.70 ± 0.01** | 0.72 ± 0.00 | **1.00 ± 0.03** | **3.0** |

are out-of-distribution. In contrast, GABO generates a candidate molecule with a long hydrocarbon backbone and *no* rings, resulting in a penalized logP score of 22.1—greater than the best observed value in the offline dataset for the task.

**Ablation Experiments.** Taking inspiration from (Trabucco et al., 2021), it is possible to utilize our SCR algorithm in GABO *without* dynamically computing $\alpha$ (and hence the Lagrange multiplier $\lambda$). To better characterize the utility of aSCR, we ablate **Algorithm 1** by treating $\lambda$ instead as a hand-tunable constant hyperparameter, and test our method using different values of $\lambda = \alpha/(1 - \alpha)$ (**Table 3**). Setting $\alpha = 0$ (i.e., $\lambda = 0$) corresponds to naïve BO against the unconstrained surrogate model, while $\alpha = 1$ (i.e., $\lambda \to \infty$) is equivalent to a WGAN-like policy. Evaluating constant values of $\alpha$ ranging from 0 to 1, we find that there is no consistently optimal constant value for all eight optimization tasks. In contrast, our method achieves an average rank of **1.9** (**2.4**) on the top-1 (top-128) evaluation metric, and is one of the top two methods when compared to the ablations for at least five of the eight tasks. These results suggest that the 'adaptive' nature of aSCR is an important component in solving the constrained optimization problem in (7).

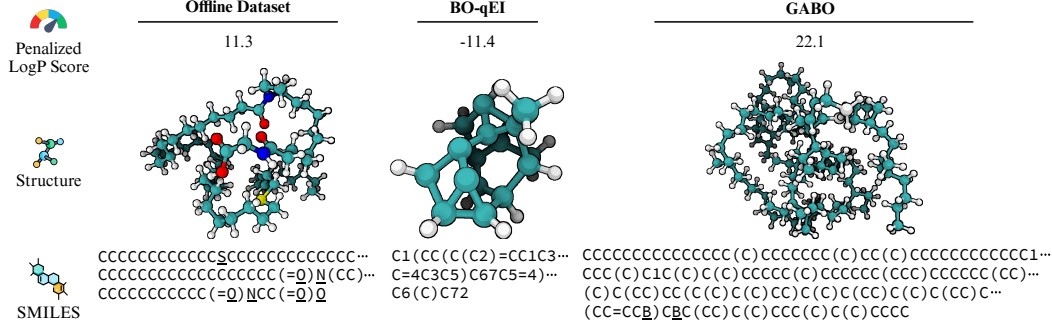

Figure 2: **Penalized LogP Score Maximization Sample Candidate Designs** (**Left**) The molecule with the highest penalized LogP score of 11.3 in the offline dataset. Separately, we show the 100th percentile candidate molecules according to the surrogate objective generated from (**Middle**) vanilla BO-qEI and (**Right**) GABO. Teal- (white-) colored atoms are carbon (hydrogen). Non-hydrocarbon atoms are underlined in the SMILES (Weininger, 1988) string representations of the molecules.

Table 3: **GABO Adaptive SCR Ablation Study** One-shot ($k = 1$) and few-shot ($k = 128$) oracle evaluations averaged across 10 random seeds reported as mean $\pm$ standard deviation. $\mathcal{D}$ (best) reports the top oracle value in the task dataset.

| Top-1 | Branin | LogP | TF-Bind-8* | GFP* | UTR* | ChEMBL* | D'Kitty | Warfarin | Rank |
|---|---|---|---|---|---|---|---|---|---|
| $\mathcal{D}$ (best) | -13.0 | 11.3 | 0.439 | 3.53 | 7.12 | 0.61 | 0.88 | -0.19 $\pm$ 1.96 | — |
| $\alpha = 0.0$ | -11.0 $\pm$ 7.8 | -52.5 $\pm$ 88.8 | 0.586 $\pm$ 0.193 | 1.43 $\pm$ 0.00 | 5.65 $\pm$ 1.30 | 0.59 $\pm$ 0.10 | 0.61 $\pm$ 0.15 | 0.16 $\pm$ 1.67 | 4.5 |
| $\alpha = 0.2$ | -9.8 $\pm$ 3.9 | -4.39 $\pm$ 60.7 | 0.535 $\pm$ 0.110 | 1.43 $\pm$ 0.00 | 4.69 $\pm$ 1.44 | 0.63 $\pm$ 0.03 | 0.61 $\pm$ 0.15 | 0.16 $\pm$ 1.79 | 3.9 |
| $\alpha = 0.5$ | -7.9 $\pm$ 6.6 | -83.9 $\pm$ 166.3 | 0.601 $\pm$ 0.212 | 1.43 $\pm$ 0.00 | 5.69 $\pm$ 1.51 | 0.63 $\pm$ 0.04 | 0.66 $\pm$ 0.12 | 0.16 $\pm$ 1.79 | 3.6 |
| $\alpha = 0.8$ | -5.2 $\pm$ 3.1 | -43.3 $\pm$ 170.0 | **0.654 $\pm$ 0.218** | 1.66 $\pm$ 0.69 | 6.49 $\pm$ 1.20 | **0.64 $\pm$ 0.02** | **0.71 $\pm$ 0.01** | 0.16 $\pm$ 1.80 | 2.4 |
| $\alpha = 1.0$ | -99.5 $\pm$ 61.2 | -46.8 $\pm$ 114.3 | 0.454 $\pm$ 0.120 | **3.74 $\pm$ 0.01** | 5.26 $\pm$ 2.35 | 0.52 $\pm$ 0.16 | 0.62 $\pm$ 0.15 | -9.04 $\pm$ 57.3 | 4.8 |
| **aSCR** | **-2.6 $\pm$ 1.1** | **21.3 $\pm$ 33.2** | 0.570 $\pm$ 0.131 | 3.60 $\pm$ 0.40 | **7.51 $\pm$ 0.39** | 0.60 $\pm$ 0.07 | **0.71 $\pm$ 0.01** | **0.60 $\pm$ 1.80** | **1.9** |

| Top-128 | Branin | LogP | TF-Bind-8* | GFP* | UTR* | ChEMBL* | D'Kitty | Warfarin | Rank |
|---|---|---|---|---|---|---|---|---|---|
| $\mathcal{D}$ (best) | -13.0 | 11.3 | 0.439 | 3.53 | 7.12 | 0.61 | 0.88 | -0.19 $\pm$ 1.96 | — |
| $\alpha = 0.0$ | **-0.4 $\pm$ 0.0** | 135.3 $\pm$ 16.0 | 0.942 $\pm$ 0.025 | 2.26 $\pm$ 1.03 | 8.26 $\pm$ 0.09 | 0.67 $\pm$ 0.00 | 0.72 $\pm$ 0.00 | 0.93 $\pm$ 0.11 | 4.3 |
| $\alpha = 0.2$ | **-0.4 $\pm$ 0.1** | 121.8 $\pm$ 20.6 | 0.925 $\pm$ 0.029 | 3.01 $\pm$ 1.04 | 8.20 $\pm$ 0.10 | 0.67 $\pm$ 0.01 | 0.72 $\pm$ 0.00 | **1.00 $\pm$ 0.00** | 4.8 |
| $\alpha = 0.5$ | **-0.4 $\pm$ 0.0** | 127.7 $\pm$ 23.1 | 0.944 $\pm$ 0.040 | 3.49 $\pm$ 0.69 | 8.29 $\pm$ 0.08 | 0.67 $\pm$ 0.01 | 0.72 $\pm$ 0.00 | **1.00 $\pm$ 0.00** | 2.9 |
| $\alpha = 0.8$ | **-0.4 $\pm$ 0.0** | 104.5 $\pm$ 31.8 | 0.933 $\pm$ 0.036 | **3.74 $\pm$ 0.00** | 8.38 $\pm$ 0.11 | 0.67 $\pm$ 0.02 | 0.72 $\pm$ 0.00 | **1.00 $\pm$ 0.00** | 3.4 |
| $\alpha = 1.0$ | -2.2 $\pm$ 1.4 | **142.3 $\pm$ 2.41** | 0.906 $\pm$ 0.061 | **3.74 $\pm$ 0.00** | **8.54 $\pm$ 0.08** | 0.68 $\pm$ 0.01 | 0.72 $\pm$ 0.00 | 0.99 $\pm$ 0.04 | 3.4 |
| **aSCR** | -0.5 $\pm$ 0.1 | 122.1 $\pm$ 20.6 | **0.954 $\pm$ 0.025** | **3.74 $\pm$ 0.00** | 8.36 $\pm$ 0.08 | **0.70 $\pm$ 0.01** | 0.72 $\pm$ 0.00 | **1.00 $\pm$ 0.03** | **2.4** |

Of note, the top designs found across different constant values of $\alpha$ can be very similar for certain tasks. This reflects the inherent challenge in developing task-agnostic methods for policy regularization—if the magnitudes of the unconstrained objective and regularization function vastly differ, then constant values of $\alpha$ may over- or under- constrain the objective. Adaptive SCR overcomes this problem by dynamically setting $\alpha$ as an implicit function of prior observations.

## 6 Conclusion

We propose **adaptive source critic regularization (aSCR)** to solve the problem of off-distribution objective evaluation in offline MBO. When leveraged with vanilla Bayesian optimization, aSCR outperforms baseline methods to achieve an average rank of **3.8** (**3.0**) in one-shot $k = 1$ (few-shot $k = 128$) oracle evaluation, and most consistently proposes designs better than the offline dataset.

**Limitations.** One limitation of aSCR is that our algorithm requires preexisting knowledge of the prior distribution over the input space in order to be computationally tractable. While we have focused our experimental evaluation on tasks amenable to imposed latent space priors, further work is needed to adapt aSCR to any arbitrary configuration space. Future work may also extend aSCR to improve parent optimization methods more sophisticated than BO-qEI and Gradient Ascent explored herein.

**Impact Statement.** Offline policy optimization methods, such as those discussed in this work, have the potential to benefit society. Such examples may include helping develop more effective drugs and individualizing patient therapies. However, as with any real-world algorithm, these methods can also be leveraged to generate potentially harmful design candidates. Careful oversight by domain experts and researchers is required to ensure that the contributions proposed herein are used for social good.

## Funding Disclosure and Acknowledgements

The authors thank Pratik Chaudhari at the University of Pennsylvania and the anonymous NeurIPS peer reviewers for their thoughtful comments, feedback, and discussion regarding this work. MSY is supported by NIH F30 MD020264. YZ and JRG are supported by NSF award IIS-2145644. JCG is supported by NIH R01 EB031722. OB is supported by NSF Award CCF-1917852.

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

# A  Additional Implementation Details

**Oracle Functions.** All oracle functions for the tasks assessed are either exact functions or approximate oracles developed by domain experts. Specifically, the **Branin** and **TF-Bind-8** tasks utilize exact oracles described in detail by Branin (1972) and Barrera et al. (2016), respectively. The oracle for the Penalized **LogP** task is an approximate oracle from Wildman & Crippen (1999) that is the same oracle used by domain experts in the Guacamol benchmarking study (Brown et al., 2019). The **GFP**, **UTR**, and **ChEMBL** tasks feature approximate oracles from Snoek et al. (2012), Angermueller et al. (2020), and Trabucco et al. (2022), respectively, that were trained on a larger, hidden datasets inaccessible to us for the respective tasks. The **D'Kitty** morphology task uses a MuJoCo (Todorov et al., 2012) simulation environment and learned control policy from Trabucco et al. (2022) to evaluate proposed designs. Finally, the **Warfarin** task uses a linear model (Consortium, 2009) to estimate a patient's optimal warfarin dose given their pharmacogenetic attributes.

**Data Preprocessing.** (1) For the **Branin** task, we sample 1000 points from the square input domain $[-5, 10] \times [0, 15]$ to construct the offline dataset, and remove the top 20%-ile according to the oracle function to make the task more challenging in line with prior work (Mashkaria et al., 2023). In this continuous task (along with the **D'Kitty** and **Warfarin** tasks), we treat input designs as their own latent space mappings, such that the VAE encoder and decoder for this task are both the identity function with zero trainable parameters. (2) The offline dataset of the Penalized **LogP** task is the validation partition of the Guacamol dataset from Brown et al. (2019), which consists of 79,564 unique molecules and their corresponding penalized LogP scores. The input molecules are represented as SMILES strings (Weininger, 1988), which is a molecule representation format shown to frequently yield invalid molecules in prior work (Krenn et al., 2020). Therefore, we encode the molecules instead as SELFIES strings, an alternative molecule representation from Krenn et al. (2020) with 100% robustness.

(3) - (5) The **TF-Bind-8**, **GFP**, and **UTR** tasks are assessed as-released by Design-Bench from Trabucco et al. (2022)—please refer to their work for task-specific descriptions. (6) - (7) In the **ChEMBL** and **D'Kitty** tasks, we normalize all objective values $y$ in the offline dataset to $\hat{y} = (y - y_{\min})/(y_{\max} - y_{\min})$ as done in prior work (Mashkaria et al., 2023), where $\hat{y}$ is the corresponding normalized objective value and $y_{\min}$ ($y_{\max}$) is the minimum (maximum) observed objective value in the full, *unobserved* dataset. Because only the bottom 60%-ile (40%-ile) from the full dataset is used in the available offline dataset for the ChEMBL (D'Kitty) task, the respective maximum $\hat{y}$ values are less than 1.0 (**Supplementary Table A1**). We also translate the original SMILES string representations in the ChEMBL task into SELFIES strings (Krenn et al., 2020) as in the LogP task.

(8) Finally, the **Warfarin** task uses the dataset of pharmacogenetic patient covariates published by Consortium (2009). We split the original dataset of 3,936 unique patient observations into training (validation) partitions with 3,736 (200) datums. The patient attributes in the Warfarin dataset consist of a combination of discrete and continuous values. All discrete attributes are one-hot encoded into binarized dimensions, and continuous values are normalized to zero mean and unit variance using the training dataset. Missing patient values were imputed following prior work (Truda & Marais, 2021). We define the cost $c(z|x)$ accrued by a patient with attributes $x \in \mathbb{R}^{32}$ as a function of the input dose $z \in \mathbb{R}$ is $c(z|x) = (z - d_{\text{oracle}}(x))^2$, where $d_{\text{oracle}} : \mathbb{R}^{32} \to \mathbb{R}$ is the domain-expert oracle warfarin dose estimator from Consortium (2009). The observed objective values $y$ associated with each of the training datums is calculated as $y = [c(\bar{z}|x) - c(z|x)]/c(\bar{z}|x)$, where $\bar{z}$ is the mean warfarin dose over the training dataset and $z$ is the true dose given to the patient. Using this constructed offline dataset, our task is then to assign optimal doses to the 200 validation patients to maximize $y$ with *no* prior warfarin dosing observations.

# B  Additional Experimental Results

In this section, we provide additional experimental results that help better characterize both the strengths and limitations of GABO and GAGA.

## B.1  How do sub-optimal design candidates proposed by GABO and GAGA perform?

To evaluate the robustness of optimization methods, we report one-shot 90th percentile oracle scores in **Supplementary Tables B1** and **B2**. For each method, all proposed designs are ranked according

Table A1: **MBO Datasets and Tasks**  Implementation details for each of the seven MBO tasks assessed in our work. *Denotes the life sciences-related discrete MBO tasks offered by the Design-Bench benchmarking repository (Trabucco et al., 2022).

| Property | Branin | LogP | TF-Bind-8* | GFP* | UTR* | ChEMBL* | D'Kitty | Warfarin |
|---|---|---|---|---|---|---|---|---|
| Dataset Size | 800 | 79,564 | 32,898 | 5,000 | 140,000 | 441 | 10,004 | 200 |
| Input Shape | 2 | 108 | 8 | 237 | 50 | 32 | 56 | 1 (33) |
| Vocab Size | — | 97 | 4 | 20 | 4 | 40 | — | — |
| VAE Backbone | Identity | Transformer | ResNet | ResNet | ResNet | Transformer | Identity | Identity |
| VAE Latent Shape | 2 | 256 | 16 | 32 | 32 | 128 | 56 | 33 |
| Oracle | Exact | Linear | Exact | Transformer | ResNet | Random Forest | Exact | Linear |
| $\mathcal{D}$ (best) | -13.0 | 11.3 | 0.439 | 3.53 | 7.12 | 0.61 | 0.88 | -0.19 ± 1.96 |

to the surrogate forward model ((8) for Generative Adversarial Bayesian Optimization (GABO) and Generative Adversarial Gradient Ascent (GAGA)), and the single 90th percentile design according to this ranking is selected and evaluated using the oracle function. We report the oracle score of this suboptimal design averaged over 10 seeds.

Our results show that GABO and GAGA do not propose suboptimal designs that are better than those proposed by other methods, such as BONET (Mashkaria et al., 2023), Simulated Annealing (Kirkpatrick et al., 1983), L-BFGS (Liu & Nocedal, 1989), and ExPT (Nguyen et al., 2023). This is not surprising, as aSCR is not designed to target this metric (and it is not our primary metric of interest). Separately for GABO, we also hypothesize that the algorithm's performance according to this metric may partially be explained by the limitations of the underlying Bayesian optimization (BO) optimization algorithm. Because BO is not an iterative first-order algorithm, the designs proposed by any BO-based algorithm often have high variance in practice—this is indeed what we observe across all of our experiments, including in **Table 1** and **Supplementary Tables B1** and **B2**.

Finally, we note that in most applications of offline optimization, the 90th percentile metric—or any metric that does not use the best proposed design(s)—is not as useful as the other metrics assessed where GABO does perform well. This is because in offline optimization tasks with a restricted budget to query the hidden, expensive-to-evaluate oracle function, we are not interested in "wasting" this limited budget on subpar design candidates. While the 90th percentile and similar metrics can be helpful to understand the limitations of algorithms (as in this case), we believe that the alternative evaluation metrics reported in the main text—namely, the 100th percentile top-1 and top-128 oracle score metrics—are more useful and practical in assessing each of the optimization algorithms.

Table B1: **Constrained Budget ($k = 1$) Suboptimal (90%-ile) Oracle Evaluation**  The oracle score of the 90th percentile design candidate according to the surrogate across 10 random seeds is reported as mean ± standard deviation. $\mathcal{D}$ (best) reports the top oracle value in the task dataset. The average rank across all seven tasks is reported in the final table column. **Bolded** (Underlined) entries indicate the best (second best) entry in the column. *Denotes the life sciences-related discrete MBO tasks from Design-Bench (Trabucco et al., 2022).

| Method | Branin | LogP | TF-Bind-8* | GFP* | UTR* | ChEMBL* | D'Kitty | Warfarin | Rank |
|---|---|---|---|---|---|---|---|---|---|
| $\mathcal{D}$ (best) | -13.0 | 11.3 | 0.439 | 3.53 | 7.12 | 0.61 | 0.88 | -0.19 ± 1.96 | — |
| Grad. | -94.4 ± 20.9 | -5.47 ± 1.32 | 0.429 ± 0.023 | 3.43 ± 0.67 | 7.16 ± 0.21 | -1.95 ± 0.00 | 0.53 ± 0.20 | 0.87 ± 1.08 | 9.1 |
| L-BFGS | **-4.0 ± 0.0** | 4.96 ± 6.64 | 0.547 ± 0.163 | 3.50 ± 0.70 | 7.36 ± 0.92 | -1.95 ± 0.00 | 0.31 ± 0.00 | 0.75 ± 1.66 | 6.9 |
| CMA-ES | -10.4 ± 3.0 | -4.35 ± 6.18 | 0.448 ± 0.068 | **3.74 ± 0.00** | 6.95 ± 1.13 | -1.95 ± 0.00 | 0.60 ± 0.29 | -4.02 ± 21.8 | 7.1 |
| Anneal | -13.2 ± 0.0 | 9.57 ± 0.66 | 0.439 ± 0.000 | 3.65 ± 0.04 | 7.41 ± 0.22 | -1.95 ± 0.00 | 0.56 ± 0.00 | **0.96 ± 0.08** | **6.0** |
| BO | -11.5 ± 2.3 | -56.2 ± 91.9 | 0.552 ± 0.152 | 1.42 ± 0.00 | 5.80 ± 1.71 | 0.64 ± 0.01 | 0.46 ± 0.18 | -36.9 ± 205 | 9.6 |
| TuRBO | -16.3 ± 10.2 | -24.3 ± 66.3 | **0.563 ± 0.087** | 1.42 ± 0.00 | 6.79 ± 1.25 | **0.65 ± 0.00** | 0.71 ± 0.01 | -32.3 ± 94.9 | 7.9 |
| BONET | -29.2 ± 2.2 | **10.8 ± 0.43** | 0.324 ± 0.041 | **3.74 ± 0.00** | **8.70 ± 0.32** | 0.56 ± 0.11 | 0.78 ± 0.00 | — | **6.0** |
| DDOM | -1870 ± 2693 | -7.10 ± 1.42 | 0.386 ± 0.224 | 1.43 ± 0.00 | 7.91 ± 0.29 | **0.65 ± 0.01** | 0.50 ± 0.19 | -56.6 ± 79.6 | 9.6 |
| COM | -3468 ± 679 | -37.4 ± 23.0 | 0.346 ± 0.093 | 3.62 ± 0.00 | 5.26 ± 1.01 | 0.60 ± 0.04 | **0.90 ± 0.01** | 0.80 ± 0.93 | 9.6 |
| RoMA | -18.5 ± 8.2 | 5.21 ± 1.39 | 0.500 ± 0.153 | 3.58 ± 0.11 | 6.94 ± 1.11 | 0.43 ± 0.18 | 0.41 ± 0.21 | -2.44 ± 2.16 | 8.1 |
| BDI | -109 ± 0.0 | 0.93 ± 0.88 | 0.471 ± 0.000 | 3.58 ± 0.05 | 5.62 ± 0.00 | 0.49 ± 0.00 | 0.76 ± 0.00 | -24.8 ± 233 | 9.0 |
| ExPT | -23.1 ± 11.3 | -16.7 ± 25.1 | 0.480 ± 0.091 | **3.74 ± 0.00** | 6.70 ± 0.39 | 0.62 ± 0.04 | 0.75 ± 0.07 | -0.40 ± 1.61 | 6.9 |
| BootGen | — | -116.8 ± 85.7 | 0.388 ± 0.007 | 3.60 ± 0.04 | 7.74 ± 0.56 | 0.61 ± 0.03 | — | — | 8.8 |
| ROMO | -3142 ± 330 | -25.6 ± 23.1 | 0.354 ± 0.247 | 3.59 ± 0.08 | 5.49 ± 1.38 | 0.62 ± 0.04 | 0.42 ± 0.17 | -2.77 ± 5.21 | 11.1 |
| **GAGA** | -14.2 ± 15.2 | -16.7 ± 81.1 | 0.546 ± 0.148 | 3.22 ± 0.86 | 6.40 ± 1.13 | -1.95 ± 0.00 | 0.89 ± 0.01 | 0.24 ± 0.20 | 8.5 |
| **GABO** | -12.7 ± 10.0 | -12.2 ± 46.1 | 0.467 ± 0.066 | 3.56 ± 1.66 | 6.12 ± 1.22 | 0.61 ± 0.08 | 0.57 ± 0.17 | 0.02 ± 5.77 | 7.9 |

Table B2: **GABO Adaptive SCR Ablation Study—Constrained Budget** ($k = 1$) **Suboptimal (90%-ile) Oracle Evaluation** The oracle score of the 90th percentile design candidate according to the surrogate across 10 random seeds reported as mean $\pm$ standard deviation. $\mathcal{D}$ (best) reports the top oracle value in the task dataset. Average method rank across all seven tasks reported in the final column. *Denotes the life sciences-related discrete MBO tasks Design-Bench (Trabucco et al., 2022).

| GABO $\alpha$ Value | Branin | LogP | TF-Bind-8* | GFP* | UTR* | ChEMBL* | D'Kitty | Warfarin | Rank |
|---|---|---|---|---|---|---|---|---|---|
| $\mathcal{D}$ (best) | -13.0 | 11.3 | 0.439 | 3.53 | 7.12 | 0.61 | 0.88 | -0.19 $\pm$ 1.96 | — |
| $\alpha = 0.0$ | -11.5 $\pm$ 2.3 | -56.2 $\pm$ 91.9 | 0.552 $\pm$ 0.152 | 1.42 $\pm$ 0.00 | 5.80 $\pm$ 1.71 | **0.64 $\pm$ 0.01** | 0.46 $\pm$ 0.18 | -36.9 $\pm$ 205 | 3.3 |
| $\alpha = 0.2$ | -9.0 $\pm$ 2.6 | -40.2 $\pm$ 77.4 | **0.612 $\pm$ 0.114** | 1.42 $\pm$ 0.00 | 5.81 $\pm$ 1.83 | 0.59 $\pm$ 0.13 | 0.49 $\pm$ 0.18 | -51.7 $\pm$ 265 | 2.9 |
| $\alpha = 0.5$ | **-8.6 $\pm$ 4.4** | -90.1 $\pm$ 107.2 | 0.501 $\pm$ 0.109 | 1.65 $\pm$ 0.69 | **6.64 $\pm$ 1.42** | 0.52 $\pm$ 0.15 | 0.41 $\pm$ 0.16 | -63.5 $\pm$ 336 | 3.9 |
| $\alpha = 0.8$ | -10.9 $\pm$ 2.1 | -41.9 $\pm$ 82.5 | 0.433 $\pm$ 0.158 | 1.97 $\pm$ 0.88 | 4.89 $\pm$ 1.23 | 0.56 $\pm$ 0.15 | 0.38 $\pm$ 0.15 | -48.5 $\pm$ 265 | 4.4 |
| $\alpha = 1.0$ | -104.6 $\pm$ 68.9 | -77.1 $\pm$ 146.1 | 0.452 $\pm$ 0.179 | 2.05 $\pm$ 0.98 | 5.15 $\pm$ 1.51 | 0.60 $\pm$ 0.08 | 0.41 $\pm$ 0.16 | -82.1 $\pm$ 552 | 4.5 |
| **aSCR** | -12.7 $\pm$ 10.0 | **-12.2 $\pm$ 46.1** | 0.467 $\pm$ 0.066 | **3.56 $\pm$ 1.66** | 6.12 $\pm$ 1.22 | 0.61 $\pm$ 0.08 | **0.57 $\pm$ 0.17** | **0.02 $\pm$ 5.77** | **2.1** |

To further characterize the distribution of designs and their associated oracle scores proposed by GABO, **Figure B1** plots a histogram of the oracle scores of (1) all 2,048 oracle scores, and (2) the oracle scores of the top 256 designs according to the penalized surrogate objective in (8) for the **LogP** task. Compared with the other optimization methods assessed, we notice that the range of oracle scores is larger for BO-based optimization methods compared with the baseline methods assessed. This helps motivate our design choice to leverage aSCR and **Algorithm 1** with BO-qEI, as BO is able to explore a larger region of the design space and is an effective parent optimizer for complex design spaces. Secondly, we also find that the distribution of scores is similar between BO-qEI and GABO, even though the performance of these two methods is remarkably different in **Tables 1** and **2**. This is likely due to the fact that while BO enables us to explore a larger effective region of the design space (compared with first-order iterative methods), **aSCR more accurately ranks proposed designs using the penalized surrogate so that we can identify promising candidates even in the low-budget oracle evaluation regime**.

## B.2 Are offline objectives and oracle function values correlated?

A key component of GABO with Adapative SCR critical to the above discussion in **Section B.1** is that generated designs score similarly according to the hidden oracle function and the regularized Lagrangian objective as in (8) in order to solve the problem of surrogate objective overestimation encountered in traditional offline optimization settings (**Fig. 1**). To assess this quantitatively, we computed the distance covariance $\mathrm{dCov}_n[\{\mathcal{L}(\mathbf{x}_k; \lambda^*)\}_{k=1}^n, \{f(\mathbf{x}_k)\}_{k=1}^n]$ between the oracle scores $f(\mathbf{x}_k)$ and the constrained Lagrangian scores $\mathcal{L}(\mathbf{x}_k; \lambda^*)$ with $\lambda = \lambda^*(t)$ computed using our Adaptive SCR algorithm. The empirical distance covariance metric is computed over the $n = 2048$ design candidates generated using our GABO algorithm. Briefly, the distance covariance is a nonnegative measure of dependence between two vectors which may be related nonlinearly; a greater distance covariance implies a greater degree of association between observations (Székely et al., 2007). We focus our subsequent discussion on the Penalized **LogP** task.

Across five random seeds, GABO with Adaptive SCR achieves a distance covariance score of 0.535 $\pm$ 0.067 (mean $\pm$ standard deviation). In contrast, naïve BO-qEI (i.e., $\lambda = 0$) only achieves a distance covariance score of 0.392 $\pm$ 0.040. Using $p < 0.05$ as a cutoff for statistical significance, the distance covariance scores are significantly different between these two methods ($p \approx 0.006$, unpaired two-tailed $t$-test). These results help support our conclusion that GABO with Adaptive SCR is able to provide better estimates of design candidate performance according to the hidden oracle function when compared to the corresponding unconstrained BO policy.

## B.3 Is adaptively computing $\alpha$ in aSCR important for the performance of GAGA?

In our ablation experiments presented in **Table 3**, we showed how that 'adaptive' nature of aSCR is an important component in solving the constrained optimization problem in (7) for GABO, and outperforms alternative approaches that manually hand-tune $\alpha$ (and hence $\lambda$) as a constant hyperparameter. We explore whether this conclusion also applies for GAGA as well here.

For clarity, we first offer the explicit formulation of GAGA in **Supplementary Algorithm 3**. We ablate **Algorithm 1** in GAGA by instead evaluating our method using different values of $\lambda = \alpha/(1 - \alpha)$. As a reminder, setting $\alpha = 0$ (i.e., $\lambda = 0$) corresponds to naïvely performing gradient

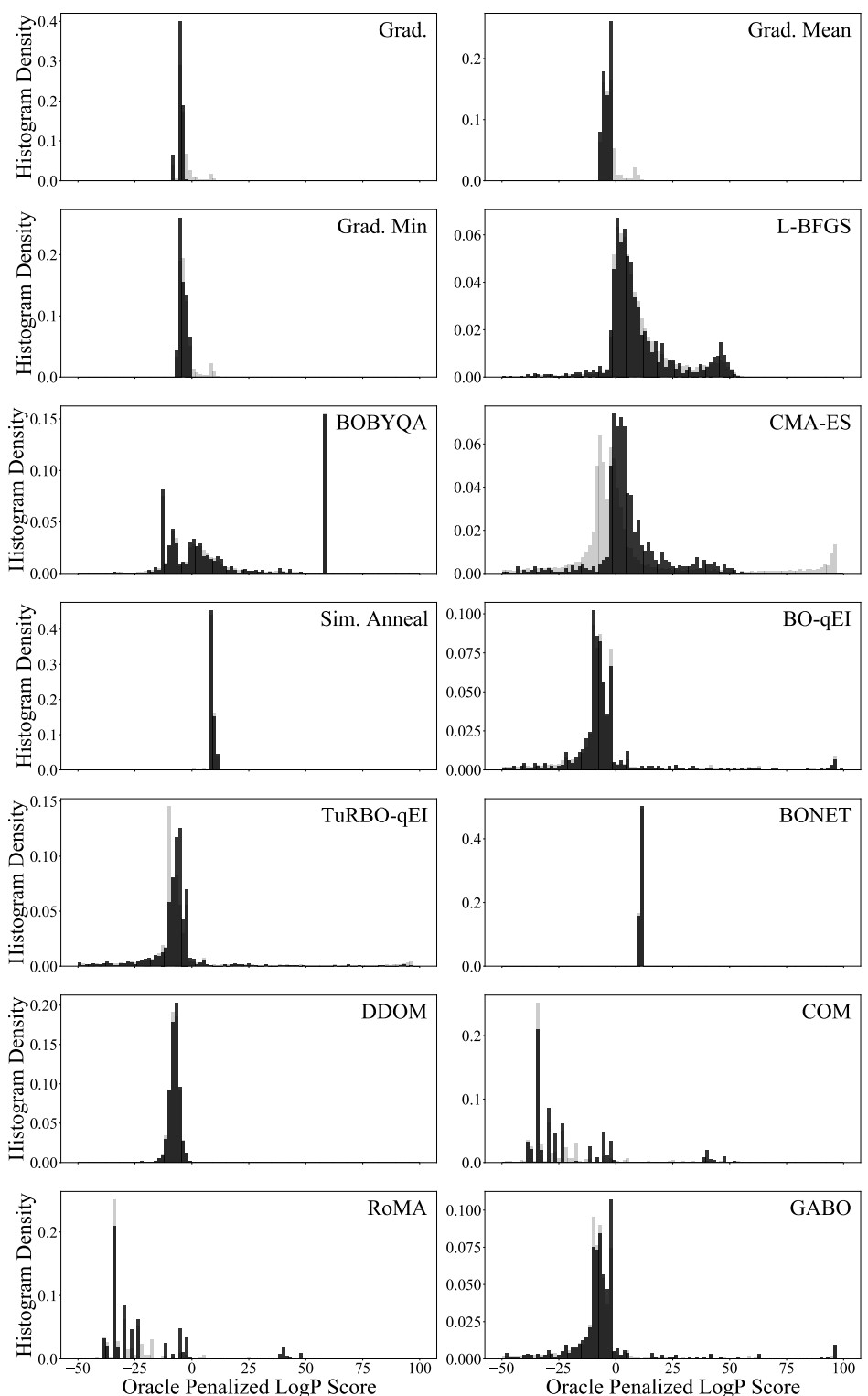

Figure B1: **Distribution of Oracle Penalized LogP Scores**   We plot the distribution of oracle scores for the top 128 surrogate model-ranked designs in black, and the distribution for all 2,048 generated designs in light gray for each of the offline model-based optimization methods assessed in our work across 10 random seeds. While GABO and BO-qEI have similar distributions, GABO is able to more reliably rank top-performing designs higher, such that these designs can be identified even under limited oracle query budgets.

ascent against the unconstrained surrogate model; setting $\alpha = 1$ (i.e., $\lambda \to \infty$) is equivalent to a WGAN-like generative policy.

---

**Algorithm 3** Generative Adversarial Gradient Ascent (GAGA)

---

**Input:** surrogate objective $f_\theta : \mathbb{R}^d \to \mathbb{R}$, offline dataset $\mathcal{D}_n = \{z_j'\}_{j=1}^n$, iterative sampling budget $T$, sampling batch size $b$, number of generator steps per source critic training $n_{\text{generator}}$, oracle query budget $k$, step size $\eta$

**AdaptiveSCR Input:** $\alpha$ step size $\Delta\alpha$, search budget $\mathcal{B}$, norm threshold $\tau$

**Define:** Differentiable source critic $c : \mathbb{R}^d \to \mathbb{R}$

**Define:** Lagrangian $\mathcal{L}(z; \alpha) : \mathbb{R}^d \times \mathbb{R} \to \mathbb{R} = -f_\theta(z) + \frac{\alpha}{1-\alpha}[\mathbb{E}_{z' \sim \mathcal{D}_n}[c(z')] - c(z)]$      // Eq. (8)

Sample $\mathcal{Z}^1 \leftarrow \{z_i^1\}_{i=1}^b$ as the top $b$ designs in $\mathcal{D}_n$ according to their previously observed oracle scores

// Train the source critic per Eq. (6) to optimality:

$c \leftarrow \text{argmax}_{||c||_L \leq K} W_1(\mathcal{D}_n, \mathcal{Z}^1) = \text{argmax}_{||c||_L \leq K} [\mathbb{E}_{z' \sim \mathcal{D}_n}[c(z')] - \mathbb{E}_{z \sim \mathcal{Z}^1}[c(z)]]$

$\alpha \leftarrow \textbf{AdaptiveSCR}(f_\theta, c, \mathcal{D}_n, \Delta\alpha, \mathcal{B}, \tau)$      // Alg. (1)

Evaluate candidates $\mathcal{Y}^1 \leftarrow \{y_i^1\}_{i=1}^b = \{-\mathcal{L}(z_i^1; \alpha)\}_{i=1}^b$

**for** $t$ in $2, 3, \ldots, T$ **do**

    $\mathcal{Z}^t \leftarrow \{z_i^t\}_{i=1}^b = \{z_i^{t-1} - \eta\nabla_{z_i^{t-1}}\mathcal{L}(z_i^{t-1}; \alpha)\}_{i=1}^b$

    $\alpha \leftarrow \textbf{AdaptiveSCR}(f_\theta, c, \mathcal{D}_n, \Delta\alpha, \mathcal{B}, \tau)$

    Evaluate samples $\mathcal{Y}^t \leftarrow \{y_i^t\}_{i=1}^b = \{-\mathcal{L}(z_i^t; \alpha)\}_{i=1}^b$

    **if** $t$ mod $n_{\text{generator}}$ equals $0$ **then**

        // Train the source critic per Eq. (6) to optimality:

        $c \leftarrow \text{argmax}_{||c||_L \leq K} W_1(\mathcal{D}_n, \mathcal{Z}^t) = \text{argmax}_{||c||_L \leq K} [\mathbb{E}_{z' \sim \mathcal{D}_n}[c(z')] - \mathbb{E}_{z \sim \mathcal{Z}^t}[c(z)]]$

    **end if**

**end for**

**return** the top $k$ samples from the $T \times b$ observations $\mathcal{D}_T = \{\{(z_i^m, y_i^m)\}_{i=1}^b\}_{m=1}^T$ according to $y_i^m$

---

Our results are shown in **Supplementary Table B3**: similar to the analogous ablation results for GABO in **Table 3**, dynamically adjusting the strength of source critic regularization using our aSCR algorithm outperforms manually setting the value of $\alpha$ to a constant in both the one-shot $k = 1$ and few-shot $k = 128$ evaluation settings.

## B.4 What is the impact of dynamic updates to the source critic over the optimization trajectory?

In **Algorithm 2** and **Supplementary Algorithm 3**, we describe how generative adversarial optimization alternates between batched acquisition steps according to the optimizer and re-training the source critic on the newly sampled trajectory points. To better interrogate the significance of dynamically re-training the source critic during optimization, we compare the performance of the default GABO and GAGA algorithms (with $n_{\text{generator}} = 4$ as the number of acquisition steps per critic retraining step) against the respective methods without source critic re-training (i.e., $n_{\text{generator}} = \infty$) in **Supplementary Table B4**. Across all three evaluation metrics and all eight tasks, dynamically retraining the source-critic improves upon the performance of the GABO when $n_{\text{generator}} = \infty$ by 67.4% in the top-1 evaluation metric; 0.0% in the top-128 evaluation metric; and 33.5% in the 90%-ile evaluation metric. Intuitively, these results align with the value of the source critic in being able to implicitly set the value of the regularization strength $\alpha$ in (8) according to the sampled trajectory points—especially in the constrained budget oracle evaluation setting.

Interestingly, we do not observe similar performance improvements with dynamic re-training of the source critic in GAGA. Qualitatively, we find that this is because of the iterative first-order nature of the parent gradient ascent algorithm—because the sampled designs are clustered in the same regions of the design space over the course of optimization, the energy landscape of the penalized surrogate (i.e., the negative of the Lagrangian expression in (8)) does not change significantly during source critic re-training. This further reinforces the optimizer to stay roughly in the same regions of the design space. As a result, it is likely that no major updates are often made to the source critic when aSCR is used in conjunction with a first-order optimization method, and so the benefit of using a finite $n_{\text{generator}}$ hyperparameter value is largely reduced when compared to its utility in GABO.

Table B3: **GAGA Adaptive ACR Ablation Study**     We ablate the dynamic computation of $\alpha$ (and hence $\lambda$ in (8)) by instead choosing to manually fix $\alpha$ to a constant value. A value of $\alpha = 0.0$ corresponds to naïve gradient ascent, and a value of $\alpha = 1.0$ corresponds to a WGAN-like generative policy. Oracle values are averaged across 10 random seeds and reported as mean $\pm$ standard deviation. In each evaluation setting, we rank all 2,048 proposed designs according to the penalized surrogate forward model in (8) and evaluate the top $k$ designs using the oracle function, reporting the maximum out of the $k$ oracle values. In the suboptimal evaluation setting, we report the oracle score of the single 90th percentile design according to the penalized surrogate ranking. **Bold** (Underlined) entries indicate the best (second best) entry in the column for the particular evaluation metric. *Denotes the life sciences MBO tasks offered by Design-Bench (Trabucco et al., 2022).

| | Branin | LogP | TF-Bind-8* | GFP* | UTR* | ChEMBL* | D'Kitty | Warfarin | Rank |
|---|---|---|---|---|---|---|---|---|---|
| $\mathcal{D}$ (best) | -13.0 | 11.3 | 0.439 | 3.53 | 7.12 | 0.61 | 0.88 | -0.19 ± 1.96 | — |
| **Constrained Budget ($k = 1$) Oracle Evaluation** | | | | | | | | | |
| $\alpha = 0.0$ | -245.1 ± 81.3 | **-5.37 ± 1.44** | 0.429 ± 0.023 | 3.18 ± 0.88 | 6.82 ± 0.21 | -1.95 ± 0.00 | 0.57 ± 0.19 | **0.86 ± 1.09** | 3.6 |
| $\alpha = 0.2$ | -13.7 ± 0.0 | -70.3 ± 115.8 | 0.439 ± 0.000 | **3.74 ± 0.00** | **7.73 ± 0.46** | -1.95 ± 0.00 | 0.88 ± 0.00 | -0.17 ± 0.00 | 2.5 |
| $\alpha = 0.5$ | -13.7 ± 0.0 | -70.3 ± 114.7 | 0.439 ± 0.000 | 3.74 ± 0.00 | 6.75 ± 0.72 | -1.95 ± 0.00 | 0.88 ± 0.00 | 0.44 ± 0.00 | 2.4 |
| $\alpha = 0.8$ | -13.7 ± 0.0 | -84.6 ± 115.8 | 0.439 ± 0.000 | 3.74 ± 0.00 | 6.75 ± 0.72 | -1.95 ± 0.00 | 0.88 ± 0.00 | 0.44 ± 0.00 | 2.6 |
| $\alpha = 1.0$ | -14.4 ± 1.5 | -27.8 ± 99.8 | 0.439 ± 0.000 | 3.74 ± 0.00 | 5.88 ± 1.04 | -1.95 ± 0.00 | **0.89 ± 0.00** | -8.61 ± 6.15 | 3.4 |
| **aSCR** | **-2.9 ± 2.2** | -68.6 ± 109.8 | **0.571 ± 0.120** | 3.74 ± 0.00 | 5.89 ± 1.42 | -1.95 ± 0.00 | **0.89 ± 0.00** | 0.01 ± 0.14 | **2.3** |
| **Relaxed Budget ($k = 128$) Oracle Evaluation** | | | | | | | | | |
| $\alpha = 0.0$ | -115.3 ± 20.8 | -5.14 ± 1.70 | **0.977 ± 0.025** | 3.49 ± 0.69 | 7.38 ± 0.15 | -1.95 ± 0.00 | 0.87 ± 0.02 | 0.86 ± 1.08 | 4.8 |
| $\alpha = 0.2$ | -13.2 ± 0.0 | 4.70 ± 10.3 | 0.439 ± 0.000 | **3.74 ± 0.00** | 7.92 ± 0.24 | -1.95 ± 0.00 | **0.95 ± 0.00** | **1.00 ± 0.00** | 2.4 |
| $\alpha = 0.5$ | -13.2 ± 0.0 | 5.07 ± 4.56 | 0.439 ± 0.000 | 3.74 ± 0.00 | 7.77 ± 0.21 | -1.95 ± 0.00 | **0.95 ± 0.01** | **1.00 ± 0.00** | 2.9 |
| $\alpha = 0.8$ | -13.2 ± 0.0 | 5.13 ± 4.28 | 0.439 ± 0.000 | 3.74 ± 0.00 | 7.44 ± 0.30 | -1.95 ± 0.00 | **0.95 ± 0.01** | **1.00 ± 0.00** | 2.9 |
| $\alpha = 1.0$ | -13.1 ± 0.0 | 5.11 ± 4.11 | 0.445 ± 0.017 | 3.74 ± 0.00 | 7.40 ± 0.28 | -1.95 ± 0.00 | 0.90 ± 0.01 | 0.96 ± 0.05 | 3.0 |
| **aSCR** | **-1.0 ± 0.2** | 14.1 ± 25.0 | 0.722 ± 0.091 | 3.74 ± 0.00 | **7.98 ± 0.36** | -1.95 ± 0.00 | 0.90 ± 0.01 | 0.95 ± 0.07 | **2.1** |
| **Constrained Budget ($k = 1$) Suboptimal (90%-ile) Oracle Evaluation** | | | | | | | | | |
| $\alpha = 0.0$ | -94.4 ± 20.9 | **-5.47 ± 1.32** | 0.429 ± 0.023 | 3.43 ± 0.67 | **7.16 ± 0.21** | -1.95 ± 0.00 | 0.53 ± 0.20 | 0.87 ± 1.08 | 3.6 |
| $\alpha = 0.2$ | -18.1 ± 0.5 | -10.9 ± 14.9 | 0.439 ± 0.000 | **3.74 ± 0.00** | 6.57 ± 0.94 | -1.95 ± 0.00 | 0.89 ± 0.02 | **0.97 ± 0.04** | 2.5 |
| $\alpha = 0.5$ | -16.2 ± 0.6 | -15.2 ± 14.4 | 0.445 ± 0.017 | 3.74 ± 0.00 | 6.75 ± 1.18 | -1.95 ± 0.00 | **0.90 ± 0.02** | 0.93 ± 0.18 | **2.4** |
| $\alpha = 0.8$ | -15.7 ± 1.0 | -12.7 ± 13.8 | 0.439 ± 0.000 | 3.74 ± 0.00 | 6.84 ± 1.29 | -1.95 ± 0.00 | 0.88 ± 0.01 | -0.24 ± 2.89 | 3.1 |
| $\alpha = 1.0$ | -14.6 ± 1.4 | -16.9 ± 13.1 | 0.439 ± 0.000 | 3.74 ± 0.00 | 6.82 ± 1.01 | -1.95 ± 0.00 | 0.89 ± 0.01 | -2.71 ± 7.71 | 3.3 |
| **aSCR** | **-14.2 ± 15.2** | -16.7 ± 81.1 | **0.546 ± 0.148** | 3.22 ± 0.86 | 6.40 ± 1.13 | -1.95 ± 0.00 | 0.89 ± 0.01 | 0.24 ± 0.20 | 3.5 |

## B.5   How does initialization affect the performance of GABO?

Per **Algorithm 2**, GABO is based on the BO-qEI baseline optimization policy, which involves initializing the gaussian process (GP) to approximate the offline surrogate model. Consistent with prior work (Eriksson et al., 2019a; Maus et al., 2022), we initialize the GP using the pseudo-random Sobol sequence (Sobol, 1967) at the beginning of the optimization procedure. However, an alternative approach is to instead initialize the GP using the top $n_{\text{init}}$ samples from the offline dataset. In particular, this strategy is already employed in both related work describing the baseline first-order optimization methods assessed herein, with the idea that better designs can be generated by initializing from better designs. We compare these two GP initialization strategies in **Supplementary Table B5**.

Interestingly, our results show that initializing the GABO GP from the Sobol sequence consistently outperforms initialization from the top candidates in offline dataset. We hypothesize that this may be due to the fact that top-scoring candidates likely lie in similar regions of the input space, which significantly alters the ability of the optimizer to explore other regions of the design space over the course of the optimization process. Future work may help better interrogate the relationship between GP initialization and offline optimization, which is outside the scope of this work.

## B.6   Can the Gaussian process (GP) in GABO be directly used as the surrogate forward model?

In **Algorithm 2**, we leverage a surrogate forward model $f_\theta$ in model-based optimization and a separate GP to acquire samples in the Bayesian optimization framework. However, it may be possible to use the GP directly as the surrogate forward model. Our results in **Supplementary Table B6** suggest that this is *not* an effective strategy with which to use GABO—using even the simple neural-network as the surrogate function (as done in our approach in **Algorithm 2**) outperforms the alternative GP-based

Table B4: **Ablating Dynamic Updates to the Source Critic**    We study the effect of training the source critic model *exactly once* (i.e., setting $n_{\text{generator}} = \infty$ in **Algorithm 2** and **Supplementary Algorithm 3**) as opposed to re-training the source critic model every $n_{\text{generator}} = 4$ acquisition steps on the newly sampled designs. Oracle values are averaged across 10 random seeds and reported as mean $\pm$ standard deviation. In each evaluation setting, we rank all 2,048 proposed designs according to the penalized surrogate forward model in (8) and evaluate the top $k$ designs using the oracle function, reporting the maximum out of the $k$ oracle values. In the suboptimal evaluation setting, we report the oracle score of the single 90th percentile design according to the penalized surrogate ranking. **Bold** entries indicate the best entry in the column for the particular optimizer and evaluation metric. *Denotes the life sciences MBO tasks offered by Design-Bench (Trabucco et al., 2022).

| GABO | Branin | LogP | TF-Bind-8* | GFP* | UTR* | ChEMBL* | D'Kitty | Warfarin |
|---|---|---|---|---|---|---|---|---|
| $\mathcal{D}$ (best) | -13.0 | 11.3 | 0.439 | 3.53 | 7.12 | 0.61 | 0.88 | $-0.19 \pm 1.96$ |
| **Constrained Budget ($k = 1$) Oracle Evaluation** | | | | | | | | |
| $n_{\text{generator}} = \infty$ | $-3.5 \pm 2.5$ | $-55.6 \pm 52.1$ | $\mathbf{0.577 \pm 0.151}$ | $\mathbf{3.74 \pm 0.00}$ | $6.73 \pm 1.10$ | $\mathbf{0.65 \pm 0.00}$ | $0.46 \pm 0.18$ | $-0.27 \pm 13.7$ |
| $n_{\text{generator}} = 4$ | $\mathbf{-2.6 \pm 1.1}$ | $\mathbf{21.3 \pm 33.2}$ | $0.570 \pm 0.131$ | $3.60 \pm 0.40$ | $\mathbf{7.51 \pm 0.39}$ | $0.60 \pm 0.07$ | $\mathbf{0.71 \pm 0.01}$ | $\mathbf{0.60 \pm 1.80}$ |
| **Relaxed Budget ($k = 128$) Oracle Evaluation** | | | | | | | | |
| $n_{\text{generator}} = \infty$ | $-0.5 \pm 0.1$ | $\mathbf{128.0 \pm 19.5}$ | $0.946 \pm 0.035$ | $3.74 \pm 0.00$ | $\mathbf{8.38 \pm 0.11}$ | $0.67 \pm 0.01$ | $0.72 \pm 0.00$ | $1.00 \pm 0.00$ |
| $n_{\text{generator}} = 4$ | $-0.5 \pm 0.1$ | $122.1 \pm 20.6$ | $\mathbf{0.954 \pm 0.025}$ | $3.74 \pm 0.00$ | $8.36 \pm 0.08$ | $\mathbf{0.70 \pm 0.01}$ | $0.72 \pm 0.00$ | $1.00 \pm 0.03$ |
| **Constrained Budget ($k = 1$) Suboptimal (90%-ile) Oracle Evaluation** | | | | | | | | |
| $n_{\text{generator}} = \infty$ | $\mathbf{-8.9 \pm 6.6}$ | $-54.1 \pm 62.6$ | $\mathbf{0.471 \pm 0.061}$ | $3.06 \pm 1.04$ | $6.02 \pm 1.41$ | $\mathbf{0.63 \pm 0.07}$ | $0.26 \pm 0.62$ | $-5.32 \pm 4.59$ |
| $n_{\text{generator}} = 4$ | $-12.7 \pm 10.0$ | $\mathbf{-12.2 \pm 46.1}$ | $0.467 \pm 0.066$ | $\mathbf{3.56 \pm 1.66}$ | $\mathbf{6.12 \pm 1.22}$ | $0.61 \pm 0.08$ | $\mathbf{0.57 \pm 0.17}$ | $\mathbf{0.02 \pm 5.77}$ |

| GAGA | Branin | LogP | TF-Bind-8* | GFP* | UTR* | ChEMBL* | D'Kitty | Warfarin |
|---|---|---|---|---|---|---|---|---|
| $\mathcal{D}$ (best) | -13.0 | 11.3 | 0.439 | 3.53 | 7.12 | 0.61 | 0.88 | $-0.19 \pm 1.96$ |
| **Constrained Budget ($k = 1$) Oracle Evaluation** | | | | | | | | |
| $n_{\text{generator}} = \infty$ | $-14.6 \pm 0.8$ | $\mathbf{-1.87 \pm 14.9}$ | $0.439 \pm 0.000$ | $3.74 \pm 0.00$ | $\mathbf{6.45 \pm 0.54}$ | $-1.95 \pm 0.00$ | $0.88 \pm 0.00$ | $-0.17 \pm 0.29$ |
| $n_{\text{generator}} = 4$ | $\mathbf{-2.9 \pm 2.2}$ | $-68.6 \pm 109.8$ | $\mathbf{0.571 \pm 0.120}$ | $3.74 \pm 0.00$ | $5.89 \pm 1.42$ | $-1.95 \pm 0.00$ | $\mathbf{0.89 \pm 0.00}$ | $\mathbf{0.01 \pm 0.14}$ |
| **Relaxed Budget ($k = 128$) Oracle Evaluation** | | | | | | | | |
| $n_{\text{generator}} = \infty$ | $-13.3 \pm 0.2$ | $\mathbf{50.2 \pm 2.48}$ | $0.439 \pm 0.000$ | $3.74 \pm 0.00$ | $7.38 \pm 0.31$ | $-1.95 \pm 0.00$ | $0.90 \pm 0.01$ | $\mathbf{0.99 \pm 0.01}$ |
| $n_{\text{generator}} = 4$ | $\mathbf{-1.0 \pm 0.2}$ | $14.1 \pm 25.0$ | $\mathbf{0.722 \pm 0.091}$ | $3.74 \pm 0.00$ | $\mathbf{7.98 \pm 0.36}$ | $-1.95 \pm 0.00$ | $0.90 \pm 0.01$ | $0.95 \pm 0.07$ |
| **Constrained Budget ($k = 1$) Suboptimal (90%-ile) Oracle Evaluation** | | | | | | | | |
| $n_{\text{generator}} = \infty$ | $-17.0 \pm 1.6$ | $\mathbf{5.88 \pm 4.88}$ | $0.439 \pm 0.000$ | $\mathbf{3.74 \pm 0.00}$ | $\mathbf{7.08 \pm 0.73}$ | $-1.95 \pm 0.00$ | $0.89 \pm 0.01$ | $-1.38 \pm 1.68$ |
| $n_{\text{generator}} = 4$ | $\mathbf{-14.2 \pm 15.2}$ | $-16.7 \pm 81.1$ | $\mathbf{0.546 \pm 0.148}$ | $3.22 \pm 0.86$ | $6.40 \pm 1.13$ | $-1.95 \pm 0.00$ | $0.89 \pm 0.01$ | $\mathbf{0.24 \pm 0.20}$ |

approach in six of the eight tasks in the top-1 evaluation setting, and is non-inferior to the alternative GP-based approach in all eight tasks in the top-128 evaluation setting. These results suggest that using a more complex neural-network surrogate function for GABO leads to better optimization results than directly using the GP as the surrogate function.

## B.7    What is the computational cost of running aSCR (i.e., Algorithm 1)?

At first glance, Adaptive SCR may appear to be a computationally expensive algorithm: it requires us to dynamically re-train a source critic neural network and compute the Lagrangian hyperparameter at each step through a grid search. However, in the implementation used for our experiments, the grid search to compute $\alpha$ is highly vectorized, and the source critic re-training patience and learning rate are such that the computational cost from re-training is not too significant. As a result, we are able to run Adaptive SCR with both Bayesian Optimization (BO) and Gradient Ascent (GA) using an experimental setup with one 24-core Intel Xeon CPU and one NVIDIA RTX A6000 GPU. To benchmark our implementation, we evaluate BO and GA both with and without our Generative Adversarial (GA) source critic regularization algorithm on the **Branin** and Penalized **LogP** optimization tasks. As a reminder, the Branin task is a standard benchmarking task for offline optimization, and the Penalized LogP task is subjectively the most challenging task assessed in our manuscript with the highest dimensional design space out of the eight assessed tasks.

Our results are shown in **Supplementary Table B7**. On the Branin toy task, aSCR increases the compute time by 257% for BO and 680% for GA, which is a significant computational cost. However, on the more challenging **LogP** task more representative of the tasks encountered in the applications of offline optimization, aSCR only introduces a 6.9% increase in compute time for GA and 28.9%

Table B5: **GABO GP Initialization Ablation Study** We investigate the effect of initializing the Gaussian process (GP) in GABO using the best $n_{\text{init}}$ points from the offline dataset (i.e., **Best** initialization strategy) versus our method in **Algorithm 2** where the GP is initialized using the first $n_{\text{init}}$ points from the Sobol sequence from (Sobol, 1967) (i.e., **Sobol** initialization strategy). Oracle values are averaged across 10 random seeds and reported as mean $\pm$ standard deviation. In each evaluation setting, we rank all 2,048 proposed designs according to the penalized surrogate forward model in (8) and evaluate the top $k$ designs using the oracle function, reporting the maximum out of the $k$ oracle values. In the suboptimal evaluation setting, we report the oracle score of the single 90th percentile design according to the penalized surrogate ranking. **Bold** entries indicate the best entry in the column for the particular optimizer and evaluation metric. *Denotes the life sciences MBO tasks offered by Design-Bench (Trabucco et al., 2022).

| Strategy | Branin | LogP | TF-Bind-8* | GFP* | UTR* | ChEMBL* | D'Kitty | Warfarin |
|---|---|---|---|---|---|---|---|---|
| $\mathcal{D}$ (best) | -13.0 | 11.3 | 0.439 | 3.53 | 7.12 | 0.61 | 0.88 | -0.19 ± 1.96 |
| | | | Constrained Budget ($k=1$) Oracle Evaluation | | | | | |
| Best | -3.6 ± 4.1 | 14.0 ± 18.4 | 0.504 ± 0.117 | 2.97 ± 1.02 | 5.36 ± 1.24 | **0.61 ± 0.00** | 0.50 ± 0.19 | -2.97 ± 9.03 |
| Sobol | **-2.6 ± 1.1** | **21.3 ± 33.2** | **0.570 ± 0.131** | **3.60 ± 0.40** | **7.51 ± 0.39** | 0.60 ± 0.07 | **0.71 ± 0.01** | **0.60 ± 1.80** |
| | | | Relaxed Budget ($k=128$) Oracle Evaluation | | | | | |
| Best | -0.5 ± 0.0 | 118.9 ± 19.5 | 0.918 ± 0.034 | 3.74 ± 0.00 | **8.37 ± 0.09** | 0.66 ± 0.01 | **0.87 ± 0.05** | 0.99 ± 0.09 |
| Sobol | -0.5 ± 0.1 | **122.1 ± 20.6** | **0.954 ± 0.025** | 3.74 ± 0.00 | 8.36 ± 0.08 | **0.70 ± 0.01** | 0.72 ± 0.00 | **1.00 ± 0.03** |
| | | | Constrained Budget ($k=1$) Suboptimal (90%-ile) Oracle Evaluation | | | | | |
| Best | **-11.8 ± 6.4** | -85.9 ± 124 | 0.382 ± 0.106 | 3.45 ± 0.77 | **6.28 ± 1.70** | 0.60 ± 0.03 | **0.64 ± 0.23** | -0.65 ± 3.97 |
| Sobol | -12.7 ± 10.0 | **-12.2 ± 46.1** | **0.467 ± 0.066** | **3.56 ± 1.66** | 6.12 ± 1.22 | **0.61 ± 0.08** | 0.57 ± 0.17 | **0.02 ± 5.77** |

Table B6: **GABO Neural Network Surrogate Ablation Study** Instead of using a neural network (NN) as our surrogate forward model, we explore if the Gaussian process (GP) employed by the parent BO optimizer can directly be used as the surrogate model in GABO's framwork. Oracle values are averaged across 10 random seeds and reported as mean $\pm$ standard deviation. In each evaluation setting, we rank all 2,048 proposed designs according to the penalized surrogate forward model in (8) and evaluate the top $k$ designs using the oracle function, reporting the maximum out of the $k$ oracle values. In the suboptimal evaluation setting, we report the oracle score of the single 90th percentile design according to the penalized surrogate ranking. **Bold** entries indicate the best entry in the column for the particular optimizer and evaluation metric. *Denotes the life sciences MBO tasks offered by Design-Bench (Trabucco et al., 2022).

| Surrogate | Branin | LogP | TF-Bind-8* | GFP* | UTR* | ChEMBL* | D'Kitty | Warfarin |
|---|---|---|---|---|---|---|---|---|
| $\mathcal{D}$ (best) | -13.0 | 11.3 | 0.439 | 3.53 | 7.12 | 0.61 | 0.88 | -0.19 ± 1.96 |
| | | | Constrained Budget ($k=1$) Oracle Evaluation | | | | | |
| GP | -37.4 ± 4.4 | -57.9 ± 159.2 | **0.576 ± 0.058** | 3.51 ± 0.69 | 6.84 ± 1.24 | **0.65 ± 0.01** | 0.42 ± 0.17 | -0.28 ± 2.13 |
| NN | **-2.6 ± 1.1** | **21.3 ± 33.2** | 0.570 ± 0.131 | **3.60 ± 0.40** | **7.51 ± 0.39** | 0.60 ± 0.07 | **0.71 ± 0.01** | **0.60 ± 1.80** |
| | | | Relaxed Budget ($k=128$) Oracle Evaluation | | | | | |
| GP | -1.5 ± 0.5 | 119.9 ± 20.1 | 0.755 ± 0.071 | 3.74 ± 0.00 | 8.34 ± 0.07 | 0.67 ± 0.01 | 0.72 ± 0.00 | -0.27 ± 2.13 |
| NN | **-0.5 ± 0.1** | **122.1 ± 20.6** | **0.954 ± 0.025** | 3.74 ± 0.00 | **8.36 ± 0.08** | **0.70 ± 0.01** | 0.72 ± 0.00 | **1.00 ± 0.03** |
| | | | Constrained Budget ($k=1$) Suboptimal (90%-ile) Oracle Evaluation | | | | | |
| GP | **-10.1 ± 10.6** | -51.5 ± 108.8 | **0.562 ± 0.091** | 2.62 ± 1.13 | **6.54 ± 1.56** | **0.65 ± 0.00** | 0.50 ± 0.19 | -0.27 ± 2.13 |
| NN | -12.7 ± 10.0 | **-12.2 ± 46.1** | 0.467 ± 0.066 | **3.56 ± 1.66** | 6.12 ± 1.22 | 0.61 ± 0.08 | **0.57 ± 0.17** | **0.02 ± 5.77** |

increase for BO. Furthermore, while there are evidently additional compute costs associated with running our aSCR algorithm, we note that in most applications of offline optimization, obtaining labeled data is the main bottleneck in many practical applications. Thus, it is often worth spending this extra compute to ensure the best results for a given evaluation budget using aSCR.

### B.8 How do the performance of GABO and other optimization methods vary with the allowed oracle query budget $k$?

To investigate this question, we vary the number of allowed $k$-shot oracle calls in the Penalized **LogP** task (**Supplementary Fig. B2**). While the majority of first-order optimization methods we evaluated

Table B7: **Computational Tractability**    Runtimes on a single node using one NVIDIA RTX A6000 GPU are averaged across 10 random seeds and reported as mean ± standard deviation.

| Time (sec) | Branin | LogP | Time (sec) | Branin | LogP |
|---|---|---|---|---|---|
| Grad. | $9.68 \pm 0.23$ | $765 \pm 6.64$ | BO | $92.1 \pm 10.2$ | $965 \pm 16.8$ |
| GAGA | $75.6 \pm 25.4$ | $818 \pm 10.5$ | GABO | $329 \pm 146$ | $1245 \pm 55.2$ |
| % Increase | $680\% \pm 259\%$ | $6.9\% \pm 1.6\%$ | % Increase | $257\% \pm 157\%$ | $28.9\% \pm 4.5\%$ |

are able to reach local optima rapidly, the proposed designs from such approaches are suboptimal compared to those from GABO (and GAGA) with Adaptive SCR as the oracle query budget size increases. Separately, comparing the curves for GABO and vanilla BO-qEI, we see that GABO with Adaptive SCR is able to propose consistently superior design candidates in the small query budget regime often encountered in real-world settings. This is due to the fact that GABO regularizes the surrogate function estimates such that the proposed candidates are both high-scoring according to the surrogate objective *and* relatively in-distribution. Our results demonstrate that especially for real-world tasks like molecule design with complex objective function landscapes, methods such as GABO with Adaptive SCR are able to explore diverse, high-performing design candidates effectively even in the setting of small oracle query budgets.

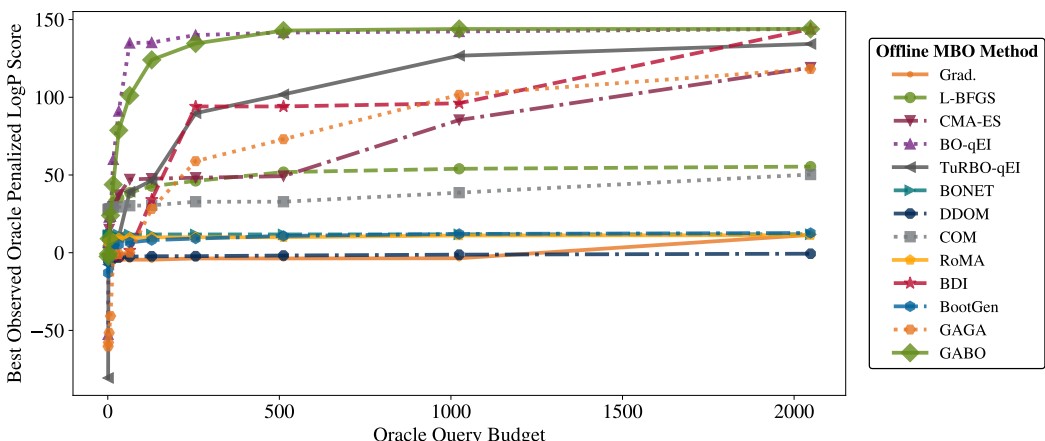

Figure B2: **100th Percentile Oracle Scores versus $k$-Shot Oracle Budget Size**    We plot the 100th percentile oracle Penalized LogP score averaged across 10 random seeds as a function of the number of allowed oracle calls $k$.

## B.9    Is the optimization budget sufficient for optimization convergence?

For all of our experimental results, we restrict the surrogate query budget to a total of 2048 allowed offline surrogate model queries in order to ensure a fair comparison between different optimization methods. To ensure that such a budget is sufficient for optimizer convergence across different optimization methods, we plot the best achieved oracle Penalized LogP value (i.e., assuming an unlimited oracle evaluation budget) as a function of the number of optimizer surrogate queries (**Supplementary Fig. B3**) for the Penalized LogP task. These results show that our methods are indeed able to converge over the course of the optimization trajectory.

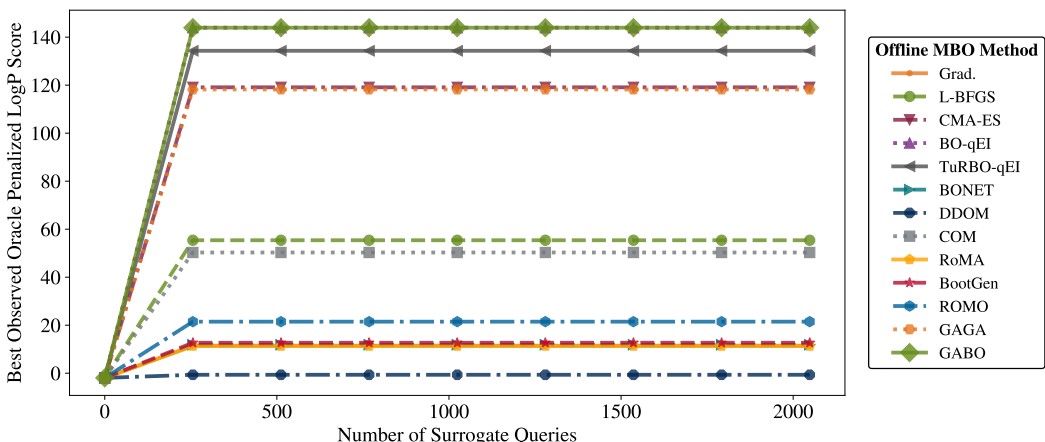

Figure B3: **Best Oracle Penalized LogP Value versus Optimization Step Count**   We plot the best Penalized LogP score averaged across 10 random seeds as a function of the number of surrogate queries made over the optimization trajectory. All offline model-based optimization (MBO) methods assessed consistently converge within the allowed oracle query budget used in our experimental setup as described in **Section 5.1**.

