# OpenReview forum: "Generative Adversarial Model-Based Optimization via Source Critic Regularization"
_NeurIPS.cc/2024/Conference — NeurIPS 2024 poster_

### Official Review · Reviewer_yPeb · 2024-07-05

**Soundness:** 3
**Presentation:** 2
**Contribution:** 2
**Rating:** 6
**Confidence:** 3

**Summary:**

The paper proposed GABO, a novel Bayesian optimization method for offline model-based optimization problems. GABO regulates the surrogate model with source critic actor so that the BO procedure remains in the in-distribution. Experiment results validate that GABO outperforms several baselines in terms of the mean rank.

**Strengths:**

- Bayesian optimization is a widely used algorithm for black-box optimization but unexplored in offline MBO settings. As far as I know, it is the first paper that improves BO for offline MBO settings

- Mathematical formulation seems to be valid, and practical algorithm is given

**Weaknesses:**

- It seems that the tasks used for evaluation from the Design Bench are discrete, **biological sequence design tasks**. There are several continuous tasks, such as Superconductor, Ant, and Dkitty, which are also high-dimensional and challenging problems. I think the research scope is then limited to offline biological sequence design, not general offline optimization tasks. Then, authors should compare their method with papers specialized in biological sequence designs, such as BIB[1] and BootGen[2].

- If my understanding is correct, the **evaluation procedure is a bit different from the offline MBO conventions**. The authors first train a surrogate model and choose the top-1 candidate among 2048 candidates according to the predicted score of the surrogate model. As the surrogate model gives inaccurate predictions on the data points outside of the offline dataset, it is not convincing to choose the best one with the surrogate model for evaluation. The authors should elaborate on the reason why they changed the evaluation setting.

[1] Chen, Can, et al. "Bidirectional learning for offline model-based biological sequence design." International Conference on Machine Learning. PMLR, 2023.

[2] Kim, Minsu, et al. "Bootstrapped training of score-conditioned generator for offline design of biological sequences." Advances in Neural Information Processing Systems 36 (2024).

**Questions:**

- In the experiment part, there is a new task called Warfarin, which is a conditional offline MBO task. I think the problem setting is practical and important. However, it seems that the proposed method lags behind other baselines in terms of performance. Could authors elaborate on why this phenomenon happens?

- It is also hard for me to accept the results of BONET and DDOM in the Branin task, as both papers deeply analyze the behavior of their methods in the Branin function. Is it due to the different evaluation settings?

**Limitations:**

As written in the weakness section, authors should specify the scope of the research and compare their method with proper baselines. Furthermore, the authors should explain why the evaluation setting has been changed.

There are a few minor comments on the manuscript.

- In the related work part, it might be beneficial that authors clearly state the limitations of prior methods rather than just explain the methods.

- In the background part, it might mislead readers if we define offline MBO as solving the optimization problem in Eq (2). There are several methods that formulate the problem as conditional generative modeling. Even for forward approaches, they do not solve the problem in Eq (2) and propose various approaches such as regularization.

---

> ### Author Rebuttal · Authors · 2024-08-07
>
> We thank Reviewer yPeb for their thoughtful comments and insights. We address their outstanding questions in our response below.
>
> **Research Scope**
>
> While we focus on the discrete, biological sequence design tasks from Design Bench, it is important to recognize that GABO and Source Critic Regularization (SCR) can be used for both continuous and discrete tasks (unlike BIB and BootGen). This is why we evaluate SCR and GABO on continuous optimization tasks in our experiments, such as the Branin and Warfarin tasks. In choosing which Design bench tasks to evaluate, we focus on the biological sequence design tasks because these tasks have been the most reproducible across recent work, as highlighted by Reviewer bMXC. The research scope is *not* limited to offline biological sequence design.
>
> Nonetheless, we evaluate BootGen on the discrete biological sequence design tasks to compare its performance against GABO for these subset of tasks. Our results are shown here:
>
> | Top $k=1$ Evaluation | LogP | TFBind | GFP | UTR | ChEMBL | Avg. Rank |
> | --- | --- | --- | --- | --- | --- | --- |
> | BootGen | -59.1 $\pm$ 69.2 | 0.398 $\pm$ 0.002 | **3.60 $\pm$ 0.03** | **7.61 $\pm$ 0.55** | **0.63 $\pm$ 0.02** | 6.3 |
> | GABO | **21.3 $\pm$ 33.2** | **0.570 $\pm$ 0.131** | **3.60 $\pm$ 0.40** | 7.51 $\pm$ 0.39 | 0.60 $\pm$ 0.07 | **4.3** |
>
> | Top $k=64$ Evaluation | LogP | TFBind | GFP | UTR | ChEMBL | Avg. Rank |
> | --- | --- | --- | --- | --- | --- | --- |
> | BootGen | 30.8 $\pm$ 14.2 | 0.401 $\pm$ 0.000 | 3.62 $\pm$ 0.00 | **8.29 $\pm$ 0.57** | 0.65 $\pm$ 0.00 | 6.6 |
> | GABO | **98.0 $\pm$ 37.6** | **0.942 $\pm$ 0.026** | **3.74 $\pm$ 0.00** | 8.25 $\pm$ 0.17 | **0.67 $\pm$ 0.02** | **2.1** |
>
> The main conclusions of our manuscript are not changed after the inclusion of BootGen as an additional baseline method. We are unable to compare GABO against BIB due to limitations in the BIB authors' publicly available code implementation.
>
> **Top-1 Experimental Evaluation**
>
> Thank you for this comment. As discussed in **Section 1**, of our manuscript, the primary motivation in evaluating the top-1 candidates is that **real-world use cases for offline generative design do not have access to large oracle query budgets**. This evaluation schema is actually studied in other related offline MBO work (e.g., [Kim et al. Proc NeurIPS 2024](https://arxiv.org/abs/2306.03111)). This evaluation schema is important because offline optimization is most helpful when evaluating newly proposed molecules requires expensive experimental laboratory setups, or when we want to optimize a patient's drug regiment without being able to test multiple doses on that patient. In these settings, the evaluation of up a large number of designs (sometimes as much 256) as in prior work is not feasible and not representative of how an algorithm would perform in practice. This change in evaluation setup is also why BONET and DDOM perform worse on the Branin task in this more realistic setting.
>
> In fact, GABO actually performs ***better*** using the more standard top-64 metric (**Supp. Tables B1 and B2**). In particular, GABO achieves an average rank of 2.1, which is better than any other reported method. Thus, our motivation for showing top-1 results is only because we believe it is more practical and relevant for real-world applications of offline model-based optimization.
>
> We also cite Reviewer bMXC's review as well for additional insights. In particular,
>   - "*One thing I really like about the paper is its evaluation which truly mirrors the offline optimization setting. The paper compares the proposed approach and baselines on a single evaluation from the oracle which I believe is the right way to evaluate algorithms for offline model-based optimization.*"
>
> **Performance on the Warfarin Task**
>
> Thank you for raising this point, and we appreciate that the Reviewer recognizes the importance of introducing practical tasks such as the Warfarin task. Compared with the other tasks assessed, the landscape of true oracle function for the Warfarin task (a LASSO model reported previously by domain experts) is uniquely a smooth convex function, and the trained surrogate likely captures similar properties over the design space. As a result, we hypothesized that this task was more conducive to first-order offline optimization methods, which is exactly what is shown in our results.
>
> **Definition of Offline MBO in Eq (2)**
>
> Thank you for this comment; we used the definition of offline MBO as in Eq (2) because (1) it is the definition of offline MBO that best frames the problem setup for our proposed approach of SCR; and (2) it is the definition consistent with the original Design-bench publication from [Trabucco et al. CoRR 2022](https://arxiv.org/abs/2202.08450). In related work that study forward approaches to offline MBO (e.g., [Yu et al. Proc NeurIPS 2021](https://arxiv.org/abs/2110.14188); [Trabucco et al. Proc ICML 2021](https://arxiv.org/abs/2107.06882); [Chen et al. Proc NeurIPS 2022](https://arxiv.org/abs/2209.07507)), the authors similarly first consider a motivating problem setup for offline MBO identical to Eq (2), and then extend on it to solve a related problem through their own methodological contributions as pointed out by the Reviewer. This is how we approached framing the problem formulation to motivate SCR as well.
>
> However, we also agree with the Reviewer that a few recent related work have used alternative formulations of offline MBO to motivate their work (e.g., [Kim et al. Proc NeurIPS 2023](https://arxiv.org/abs/2306.03111); [Krishnamoorthy et al. Proc ICML 2023](https://arxiv.org/abs/2206.10786); [Krishnamoorthy et al. Proc ICML 2023](https://arxiv.org/abs/2306.07180)). In the final manuscript, we will be sure to clarify that (1) such other methods for offline MBO exist; (2) we consider one possible definition of offline MBO in Eq (2); and (3) how these additional problem formulations relate to Eq (2) considered in our work.

---

> ### Comment · Reviewer_yPeb · 2024-08-09
>
> Thank you for your detailed response. Unfortunately, several questions remain on the paper. Here are some follow-up questions.
>
> **Research Scope)**
>
> While authors conduct continuous tasks, Branin and Warfarin, those tasks are close to toy experiment settings. Branin is a 2D function, and Warfarin is a smooth convex function. While the superconductor task in Design-Bench has a reproducibility issue, other tasks, such as Ant and Dkitty, do not suffer from that issue.
>
> Furthermore, it seems the results of BootGen significantly deviate from the reported score. It is hard for me to accept the results that the maximum score of $k=64$ in TFBind8 is 0.401, which is even below the maximum of the dataset. As shown in the Figure 4.1 of BootGen, the performance of even K=1 seems higher than 0.8.
>
> **Top-1 Experimental Evaluation)**
>
> While I also agree with the motivation that we do not have large oracle query budgets, evaluating top-1 candidates may lead to high variance. Furthermore, using a proxy to select one candidate is also not convincing as a proxy is fragile to out-of-distribution errors.
>
> I strongly recommend that authors compare the proposed method with a commonly used evaluation setting, sample 128 designs without filtering with proxy (I think this part is also a procedure of the proposed algorithm and should not be included in baselines), and report the maximum and median scores.
>
> **Performance on the Warfarin Task)**
>
> The authors say that the oracle function of Warfarin is a convex function. However, the motivation of offline MBO is optimizing high-dimensional and complex black-box functions. I think the Warfarin task may not be appropriate for evaluating MBO algorithms.
>
> Furthermore, it may be appreciated if the proposed algorithm is compared with [1], which also deals with conditional MBO problems.
>
> [1] Chen, Mingcheng, et al. "ROMO: Retrieval-enhanced Offline Model-based Optimization." Proceedings of the Fifth International Conference on Distributed Artificial Intelligence. 2023.

---

> > ### Author Response · Authors · 2024-08-11
> >
> > We thank Reviewer yPeb for their continued feedback and discussion of our work. Please find our responses to the questions below.
> >
> > **On the difference in our results compared with those reported by the BootGen authors:** This difference is because the authors normalize their reported scores and report $z=\frac{y-y_{\text{min}}}{y_{\text{max}}-y_{\text{min}}}$, where $y_{\text{min}}$ and $y_{\text{max}}$ are the worst and best scores in the offline dataset, respectively. In contrast, we report the unnormalized score $y$ in our work. For the TFBind task, an unnormalized score of $y=0.398$ (as reported by our experiments above) corresponds a normalized score of $z\approx (0.398 - 0.0)/(0.439 - 0.0)\approx 0.906$, which is aligned with expected results as reported in the original BootGen paper.
> >
> > **On the inclusion of experimental results using the Ant and D'Kitty datasets:** We are happy to evaluate our method on the D'Kitty task:
> >
> > | Method | D'Kitty Score (Top-1 Evaluations) | Final Average Rank |
> > | --- | --- | --- |
> > | $\mathcal{D}(\text{best})$ | 199 | N/A |
> > | Grad. Ascent | -185 $\pm$ 228 | 11.0 |
> > | L-BFGS | -504 $\pm$ 0 | 9.4 |
> > | CMA-ES | -503 $\pm$ 0.7 | 11.8 |
> > | Sim. Anneal | -204 $\pm$ 216 | 7.3 |
> > | BO-qEI | -140 $\pm$ 184 | 9.9 |
> > | TuRBO-qEI | -343 $\pm$ 225 | 10.0 |
> > | BONET | 74.6 $\pm$ 0.0 | 5.9 |
> > | DDOM | -501 $\pm$ 97.4 | 12.8 |
> > | COM | 218 $\pm$ 20.4 | 6.8 |
> > | RoMA | -517 $\pm$ 327 | 10.5 |
> > | BDI | -59.0 $\pm$ 0.0 | 12.5 |
> > | ExPT | -70.7 $\pm$ 239 | 10.3 |
> > | ROMO | -96.1 $\pm$ 332 | 11.3 |
> > | **GABO** | -10.2 $\pm$ 9.9 | 4.4 |
> >
> > | Method | D'Kitty Score (Top-64 Evaluations) | Final Average Rank |
> > | --- | --- | --- |
> > | $\mathcal{D}(\text{best})$ | 199 | N/A |
> > | Grad. Ascent | -127 $\pm$ 206 | 14.4 |
> > | L-BFGS | -504 $\pm$ 0 | 11.9 |
> > | CMA-ES | 199 $\pm$ 0.0 | 10.0 |
> > | Sim. Anneal | 199 $\pm$ 0.0 | 9.1 |
> > | BO-qEI | -4.1 $\pm$ 7.5 | 5.7 |
> > | TuRBO-qEI | -1.4 $\pm$ 0.6 | 6.4 |
> > | BONET | 78.1 $\pm$ 0.0 | 6.7 |
> > | DDOM | 3.4 $\pm$ 1.6 | 9.1 |
> > | COM | 230 $\pm$ 22.0 | 10.7 |
> > | RoMA | -225 $\pm$ 68 | 8.7 |
> > | BDI | -59.0 $\pm$ 0.0 | 12.9 |
> > | ExPT | 76.6 $\pm$ 65.2 | 8.7 |
> > | ROMO | 191 $\pm$ 43.6 | 14.3 |
> > | **GABO** | -2.2 $\pm$ 3.9 | 2.1 |
> >
> > After including all 8 offline MBO tasks, GABO still ranks as the top method according to both the top-1 and top-64 evaluation metrics. These new experimental results further support the utility of GABO and SCR across a wide variety of different domains.
> >
> > Regarding the Ant task, we cite [this GitHub issues link](https://github.com/brandontrabucco/design-bench/issues/23) describing a critical issue of the Ant task that we replicated in our own experiments. For this reason, we argue that it would be best to include only results on the D'Kitty task instead of the Ant task, although are happy to discuss further with the Reviewer as needed.
> >
> > **On the high variance of top-1 candidate evaluation:** We run 10 random seeds to reduce variance of the top-1 estimate. Importantly, running the top-1 sample multiple times with different random seeds is very different from taking the top-k from one random seed; in fact, the latter is potentially still a high variance evaluation since the $k$ different samples can be highly correlated. Furthermore, the fact that the top-1 value is high-variance a *very important reason* to present top-1 candidates with multiple random seeds instead of top-k results, as the use of top-k metrics for $k\gg1$ can misleadingly hide this variability.
> >
> > Finally, we note that we have included the top-64 candidate evaluations in **Supplementary Tables B1 and B2**, which would be useful for readers that may be interested in settings with a potentially larger oracle query budget. In addition, as requested by the Reviewer, we have also run similar results for top-128; GABO again performs by far the best:
> >
> > | Method | Top-128 Evaluation |
> > | --- | --- |
> > | Grad. | 15.3 |
> > | Grad. Mean. | 13.9 |
> > | Grad. Min. | 13.6 |
> > | L-BFGS | 10.7 |
> > | BOBYQA | 13.6 |
> > | CMA-ES | 11.1 |
> > | Sim. Anneal | 9.4 |
> > | BO-qEI | 5.6 |
> > | TuRBO-qEI | 6.3 |
> > | BONET | 7.8 |
> > | DDOM | 9.7 |
> > | COM | 12.0 |
> > | RoMA | 8.9 |
> > | BDI | 14.0 |
> > | ExPT | 9.7 |
> > | BootGen | 6.0 |
> > | ROMO | 14.7 |
> > | Ours | 2.7 |
> >
> > We will add these additional experimental results to our final manuscript.

---

> > > ### Author Response · Authors · 2024-08-11
> > >
> > > **On using a proxy surrogate function to select one candidate:** First, we note that using the proxy surrogate function to rank and select candidates has been used extensively in virtually all prior work that use a proxy forward predictive model for offline MBO (i.e., [Yu et al. Proc NeurIPS 2021](https://proceedings.neurips.cc/paper/2021/hash/24b43fb034a10d78bec71274033b4096-Abstract.html); [Trabucco et al. Proc ICML 2021](https://proceedings.mlr.press/v139/trabucco21a.html); [Fu et al. Proc ICLR 2021](https://arxiv.org/pdf/2102.07970)).
> > >
> > > In addition, we agree that naive selection of the design using just the function $f_{\theta}$ is a poor strategy in offline MBO due to out-of-distribution errors. This is why in our method, GABO and GAGA select the optimal designs according to the ***penalized surrogate***, i.e. the Lagrangian $\mathcal{L}(\mathbf{x}; \lambda)$ shown in Equation (8) on page 4 of our manuscript. In this way, proposed designs that are wildly out-of-distribution would have a lower penalized surrogate score, which helps make sampling based on the Lagrangian proxy more robust. Such a "sampling via regularized proxy approach" is also employed by other offline MBO works (e.g., [Trabucco et al. Proc ICML 2021](https://proceedings.mlr.press/v139/trabucco21a.html)). We will be sure to better highlight this feature of our approach in our final manuscript.
> > >
> > > Finally, as mentioned above, we have included top-64 results in our manuscript (and will add top-128 results as shown above), and GABO achieves very strong performance on this metric (in fact, stronger than top-1 performance).
> > >
> > > **On evaluating baselines without filtering via the proxy:** We clarify that for each baseline, we have used the strategy implemented by that baseline to choose the top-k candidates. In particular, the following baselines use the proxy to rank and select candidates: [Yu et al. Proc NeurIPS 2021](https://proceedings.neurips.cc/paper/2021/hash/24b43fb034a10d78bec71274033b4096-Abstract.html); [Trabucco et al. Proc ICML 2021](https://proceedings.mlr.press/v139/trabucco21a.html); [Fu et al. Proc ICLR 2021](https://arxiv.org/pdf/2102.07970). And, the following baselines use the last $k$ samples: [Mashkaria et al. Proc ICML 2023](https://proceedings.mlr.press/v202/mashkaria23a.html); [Krishnamoorthy et al. Proc ICML 2023](https://proceedings.mlr.press/v202/krishnamoorthy23a); [Nguyen et al. Proc NeurIPS 2023](https://arxiv.org/abs/2310.19961).
> > >
> > > **On the inclusion of the Warfarin Task as a new offline MBO task:** We thank the Reviewer for this comment. We note that **the key motivation of offline MBO is optimizing expensive-to-evaluate black-box functions with real-world significance**. In many cases, such real-world functions are high-dimensional and non-convex as noted by the reviewer. However, this is by no means a *requirement* for offline MBO to be useful -- as demonstrated by the Warfarin task, real-world tasks can also have convex objectives that are *just as important* for offline MBO methods to perform well on. Many prior works on offline MBO have exclusively focused on the evaluating nonconvex real-world tasks, although convex real-world objectives also have important utility in evaluating the real-world significance of methods such as GABO.
> > >
> > > **On including ROMO as a baseline method:** **Across all 8 tasks, ROMO has an average rank of 11.3 on the top-1 evaluation; 14.3 on the top-64 evaluation; and 14.7 on the top-128 evaluation.** Of specific interest is how ROMO compares with GABO on the Warfarin task, which is an example of a constrained MBO (CoMBO) problem of the kind specifically tackled by ROMO:
> > >
> > > | Warfarin Task | Top-1 Evaluation | Top-64 Evaluation | Top-128 Evaluation |
> > > | --- | --- | --- | --- |
> > > | ROMO | -0.71 $\pm$ 2.10 | -0.27 $\pm$ 0.55 | 0.76 $\pm$ 1.91 |
> > > | GABO | 0.60 $\pm$ 1.80 | 1.00 $\pm$ 0.28 | 1.00 $\pm$ 0.03 |
> > >
> > > GABO outperforms ROMO on the above evaluation metrics. Furthermore, based on the method rankings across the 8 tasks, ROMO evidently struggles on general, unconstrained MBO tasks (i.e., all tasks assessed in our work other than the Warfarin task). In contrast, our method (SCR) is not specifically designed for such problems, and is a task- and optimizer-agnostic framework for tackling a wide variety of continuous, discrete, and constrained MBO problems.
> > >
> > > We thank Reviewer yPeb again for their continued engagement and interest in our work! We would be happy to answer any additional questions they might have.

---

> > > > ### Comment · Reviewer_yPeb · 2024-08-11
> > > >
> > > > Thank you for your detailed response to my feedback. Here are some further follow-up questions for clarifications.
> > > >
> > > > **On the difference in our results compared with those reported by the BootGen authors**: BootGen authors normalize their scores with the worst score and the best score of the whole dataset (in the TFBind8 task, all possible combinations), not the maximum score of the offline dataset. You can see in Table 4.1 that while the maximum score of the offline dataset is 0.439, BootGen and other baselines achieve higher scores than the offline dataset.
> > > >
> > > > **On the inclusion of experimental results using the Ant and D'Kitty datasets**: Thank you for your additional experiments. I find that the GABO does not perform well in the DKitty task, especially with $k=64$, while still achieving the best score in terms of the mean rank. Could authors elaborate on why such a phenomenon happens? I acknowledged that it is extremely hard to achieve the best results on every design-bench task. I just wonder why such a phenomenon happens.
> > > >
> > > > **On the high variance of top-1 candidate evaluation**: The conventional evaluation protocol of Design-Bench with 8 random seeds with $k=128$ candidates, not a single seed. In my humble opinion, evaluating $k=1$ with 10 random seeds has a higher variance compared to $k=128$ with 8 random seeds.
> > > >
> > > > I appreciate that the authors include the $k=128$ results. By the way, does the evaluation still have a filtering procedure? Furthermore, it would be nice to show the overall performance, not just the average ranking.
> > > >
> > > > **On using a proxy surrogate function to select one candidate**: Thank you for the clarification that authors use a penalized surrogate instead of naive MLP for filtering. However, as far as I know, there are now ranking and filtering procedures in the paper you mentioned: RoMA, COMs, and NEMO. As far as I know, those methods perform gradient ascent to select candidates and do not have filtering procedures.
> > > >
> > > > **On evaluating baselines without filtering via the proxy**: Could authors elaborate on which procedure is used for choosing the top-k candidates from proxy in RoMA, COMs, and NEMO? Could authors also elaborate on what does it mean the "last k samples"  for BONET, DDOM, and ExPT? As far as I know, BONET and ExPT generate four trajectories with a predictive length of 64, which results in 256 samples, and DDOM generates 256 candidates using a conditional diffusion model.
> > > >
> > > > **On the inclusion of the Warfarin Task as a new offline MBO task**: I'm still not convinced well of the claim that real-world tasks can also have convex objectives as they are extremely rare. Furthermore, the performance of the proposed method in the Warfarin task is relatively low. I think it might be beneficial to find a task that can show the capability of the proposed method.
> > > >
> > > > **On including ROMO as a baseline method**: I appreciate that the authors include the comparison with ROMO. It would be beneficial to include the comparison with ROMO in the manuscript.

---

> > > > > ### Author Response · Authors · 2024-08-12
> > > > >
> > > > > Thank you for your follow-up questions! Please find our responses below:
> > > > >
> > > > > **On the difference in our results compared with those reported by the BootGen authors:** We apologize for the initial confusion; indeed, we notice that the BootGen authors use a different normalization schema as highlighted by the Reviewer - we appreciate the Reviewer for bringing this to our attention. We have adjusted our evaluation code script to account for this difference, and are happy to report the updated results for top-1, top-64, and top-128 evaluation, specifically focusing on the discrete biological sequence tasks for which BootGen is designed for:
> > > > >
> > > > > | Top-1 Evaluation | Molecule | TFBind-8 | GFP | UTR | ChEMBL | Average Rank |
> > > > > | --- | --- | --- | --- | --- | --- | --- |
> > > > > | BootGen | -13.0 $\pm$ 15.1 | 0.942 $\pm$ 0.022 | 3.10 $\pm$ 0.74 | 8.30 $\pm$ 0.93 | 0.59 $\pm$ 0.07 | 5.3 |
> > > > > | GABO | 21.2 $\pm$ 33.2 | 0.570 $\pm$ 0.131 | 3.60 $\pm$ 0.40 | 7.51 $\pm$ 0.39 | 0.60 $\pm$ 0.07 | 4.3 |
> > > > >
> > > > > | Top-64 Evaluation | Molecule | TFBind-8 | GFP | UTR | ChEMBL | Average Rank |
> > > > > | --- | --- | --- | --- | --- | --- | --- |
> > > > > | BootGen | 6.6 $\pm$ 3.1 | 0.978 $\pm$ 0.002 | 3.73 $\pm$ 0.00 | 10.5 $\pm$ 0.96 | 0.68 $\pm$ 0.01 | 2.7 |
> > > > > | GABO | 98.0 $\pm$ 37.6 | 0.942 $\pm$ 0.026 | 3.74 $\pm$ 0.00 | 8.25 $\pm$ 0.17 | 0.67 $\pm$ 0.02 | 2.4 |
> > > > >
> > > > > | Top-128 Evaluation | Molecule | TFBind-8 | GFP | UTR | ChEMBL | Average Rank |
> > > > > | --- | --- | --- | --- | --- | --- | --- |
> > > > > | BootGen | 8.1 $\pm$ 3.3 | 0.979 $\pm$ 0.002 | 3.74 $\pm$ 0.00 | 10.5 $\pm$ 0.95 | 0.69 $\pm$ 0.00 | 3.3 |
> > > > > | GABO | 122.1 $\pm$ 20.6 | 0.954 $\pm$ 0.025 | 3.74 $\pm$ 0.00 | 8.36 $\pm$ 0.08 | 0.70 $\pm$ 0.01 | 3.0 |
> > > > >
> > > > > We have verified that our top-128 results on the TFBind-8, GFP, and UTR tasks agree with those reported by the BootGen authors (accounting for their change in normalization schema). The average rank is computed across all 8 tasks for GABO, and across only the 5 discrete sequence design tasks for BootGen reported above.
> > > > >
> > > > > As the results indicate above, GABO is a competitive method with BootGen on discrete biological sequence design tasks. Considering that BootGen is proposed primarily for biological sequence design whereas GABO is general purpose, we believe these results demonstrate the effectiveness of GABO.
> > > > >
> > > > > **On the inclusion of experimental results using the Ant and D'Kitty datasets:** We thank the Reviewer for raising this point: indeed, it is challenging for *any* MBO method to achieve the best results on *every* task. We had looked into GABO's performance on the D'Kitty task and hypothesize that it may struggle due to two reasons: (1) the surrogate model and (2) the base BO optimizer for GABO:
> > > > >   1. Recall that in our evaluation setup, we use the same task-agnostic surrogate model (FCNN with 2 hidden layers) trained with task-agnostic hyperparameters. We chose such an evaluation schema because we are interested in evaluating GABO across a wide variety of settings where the surrogate model may sometimes fit the oracle well (resulting in better relative performance); and other times may not fit the oracle well (resulting in worse performance). Based on our experiments, we found our surrogate model to be limited in its expressivity of the complex design landscape of the D'Kitty task. We believe that the performance of GABO may be improved by using it with a more intelligent, task-specific surrogate model that is specifically adapted for tasks such as the D'Kitty task. The challenge of identifying and implementing such a surrogate model is left as future work.
> > > > >   2. In our experiments, we found that other BO-based baselines (i.e., BO-qEI and TuRBO-qEI) also underperform relative to other optimization methods on the D'Kitty task. One contributory component to GABO's D'Kitty performance may therefore be that BO-based optimization methods are ill-suited for this task. This is further supported by the fact that GAGA performs better than GABO on this task (see the table of results below).
> > > > >
> > > > > **On the high variance of top-1 candidate evaluation:** We very much agree with the Reviewer's comment: it is expected that evaluating with $k=1$ with 10 random seeds has a higher variance compared to $k=128$ with a similar number random seeds. The reason why we are interested in reporting this $k=1$ evaluation and capturing this expected increase in variance in our results is highlighted in our earlier comment: specifically, we (alongside Reviewer bMXC) believe that the $k=1$ evaluation is the most realistic, real-world evaluation method for offline MBO tasks.

---

> > > > > > ### Author Response · Authors · 2024-08-12
> > > > > >
> > > > > > We would also be happy to share the the full table on each of the tasks for **top-128 evaluation** (which will also be included in the final manuscript):
> > > > > >
> > > > > > | Method | Branin | LogP | TFBind8 | GFP | UTR | ChEMBL | Warfarin | D'Kitty	| Average Rank |
> > > > > > | --- | --- | --- | --- | --- | --- | --- | --- | --- | --- |
> > > > > > | $\mathcal{D}$(best) | -13.0 | 11.3 | 0.439 | 3.525 | 7.123 | 0.605 | -0.194 $\pm$ 1.957 | 0.88 | - |
> > > > > > | Grad. | -115.3 $\pm$ 20.8 | -5.1 $\pm$ 1.7 | 0.977 $\pm$ 0.025 | 3.49 $\pm$ 0.69 | 7.38 $\pm$ 0.15 | -1.95 $\pm$ 0.00 | 0.86 $\pm$ 1.08 | 0.87 $\pm$ 0.02 | 11.0 |
> > > > > > | L-BFGS | -4.0 $\pm$ 0.0 | 42.8 $\pm$ 9.4 | 0.633 $\pm$ 0.140 | 3.74 $\pm$ 0.00 | 7.51 $\pm$ 0.39 | -1.95 $\pm$ 0.00 | 0.75 $\pm$ 1.67 | 0.31 $\pm$ 0.0 | 10.1 |
> > > > > > | CMA-ES | -4.3 $\pm$ 1.7 | 47.6 $\pm$ 5.5 | 0.810 $\pm$ 0.235 | 3.74 $\pm$ 0.00 | 7.40 $\pm$ 0.32 | -1.95 $\pm$ 0.00 | -8.62 $\pm$ 63.8 | 0.74 $\pm$ 0.00 | 9.8 |
> > > > > > | Sim. Anneal | -7.4 $\pm$ 2.8 | 11.3 $\pm$ 0.0 | 0.890 $\pm$ 0.035 | 3.72 $\pm$ 0.00 | 7.96 $\pm$ 0.22 | -1.95 $\pm$ 0.00 | 0.97 $\pm$ 0.08 | 0.88 $\pm$ 0.00 | 9.3 |
> > > > > > | BO-qEI | -0.4 $\pm$ 0.0 | 135.3 $\pm$ 16.0 | 0.942 $\pm$ 0.0254 | 2.26 $\pm$ 1.03 | 8.26 $\pm$ 0.09 | 0.67 $\pm$ 0.00 | 0.93 $\pm$ 0.11 | 0.72 $\pm$ 0.00 | 6.6 |
> > > > > > | TuRBO-qEI | -0.7 $\pm$ 0.4| 59.7 $\pm$ 51.3 | 0.895 $\pm$ 0.049 | 1.89 $\pm$ 0.92 | 8.26 $\pm$ 0.11 | 0.67 $\pm$ 0.01 | 0.99 $\pm$ 0.01 | 0.72 $\pm$ 0.00 | 7.4 |
> > > > > > | BONET | -26.0 $\pm$ 0.9 | 11.7 $\pm$ 0.4 | 0.951 $\pm$ 0.035 | 3.74 $\pm$ 0.00 | 9.13 $\pm$ 0.08 | 0.67 $\pm$ 0.01 | --- | 0.95 $\pm$ 0.01 | 5.6 |
> > > > > > | DDOM | -18.4 $\pm$ 29.8 | -2.2 $\pm$ 0.6 | 0.936 $\pm$ 0.051 | 1.44 $\pm$ 0.00 | 8.29 $\pm$ 0.33 | 0.66 $\pm$ 0.01 | 1.00 $\pm$ 0.00 | 0.89 $\pm$ 0.01 | 8.4 |
> > > > > > | COM | -1981.0 $\pm$ 224.5 | 45.0 $\pm$ 17.0 | 0.902 $\pm$ 0.056 | 3.62 $\pm$ 0.00 | 8.18 $\pm$ 0.00 | 0.64 $\pm$ 0.01 | 0.77 $\pm$ 0.86 | 0.95 $\pm$ 0.02 | 8.5 |
> > > > > > | RoMA | -4.8 $\pm$ 3.0 | 10.8 $\pm$ 0.8 | 0.760 $\pm$ 0.113 | 3.74 $\pm$ 0.00 | 8.12 $\pm$ 0.09 | 0.69 $\pm$ 0.03 | 0.67 $\pm$ 0.05 | 1.02 $\pm$ 0.04 | 7.8 |
> > > > > > | BDI | -65.0 $\pm$ 51.3 | 1.5 $\pm$ 5.8 | 0.735 $\pm$ 0.086 | 3.61 $\pm$ 0.05 | 6.31 $\pm$ 0.00 | 0.50 $\pm$ 0.12 | -5.07 $\pm$ 21.0 | 0.94 $\pm$ 0.00 | 11.8 |
> > > > > > | ExPT | -1.7 $\pm$ 1.0 | -6.5 $\pm$ 4.6 | 0.927 $\pm$ 0.095 | 3.74 $\pm$ 0.00 | 8.12 $\pm$ 0.09 | 0.68 $\pm$ 0.04 | 0.96 $\pm$ 0.05 | 0.97 $\pm$ 0.01 | 6.5 |
> > > > > > | BootGen | --- | 8.1 $\pm$ 3.3 | 0.979 $\pm$ 0.002 | 3.74 $\pm$ 0.00 | 10.5 $\pm$ 0.95 | 0.68 $\pm$ 0.00 | --- | --- | 3.3 |
> > > > > > | ROMO | -2367.3 $\pm$ 787.5 | -6.0 $\pm$ 14.5 | 0.572 $\pm$ 0.202 | 3.67 $\pm$ 0.03 | 6.94 $\pm$ 1.07 | 0.66 $\pm$ 0.00 | 0.76 $\pm$ 1.91 | 0.90 $\pm$ 0.02 | 12.1 |
> > > > > > | **GAGA (Ours)** | -1.0 $\pm$ 0.2 | 14.1 $\pm$ 25.0 | 0.722 $\pm$ 0.091 | 3.74 $\pm$ 0.00 | 7.98 $\pm$ 0.36 | -1.95 $\pm$ 0.00 | 0.95 $\pm$ 0.07 | 0.902 $\pm$ 0.01 | 7.6 |
> > > > > > | **GABO (Ours)** | -0.5 $\pm$ 0.1 | 122.1 $\pm$ 20.6 | 0.954 $\pm$ 0.025 | 3.74 $\pm$ 0.00 | 8.36 $\pm$ 0.08 | 0.70 $\pm$ 0.01 | 1.00 $\pm$ 0.03 | 0.72 $\pm$ 0.00 | **3.0** |
> > > > > >
> > > > > > GABO performs best on 3 tasks and second best on 2 tasks in our testing suite, with a final rank of **3.0** (best out of all assessed methods) on this evaluation metric after making the adjustments to our BootGen evaluation schema mentioned above. Furthermore, GAGA (i.e., SCR with Gradient Ascent instead of BO) also improves upon vanilla Gradient Ascent.
> > > > > >
> > > > > > Crucially, we note that this evaluation does *not* have a filtering procedure (discussed in detail below).
> > > > > >
> > > > > > **On the discussion of filtering using the proxy surrogate function**: Apologies, we believe that we misunderstood the earlier comment from the Reviewer. Notably, the BootGen work from [Kim et al. Proc NeurIPS 2023.](https://arxiv.org/abs/2306.03111) features a discussion of what it means to filter designs from the generative process using a proxy function that is different than what we meant by filtering in previous comments.
> > > > > >
> > > > > > According to their definition, GABO does not and has not previously used filtering via proxy. The same goes with the other baseline optimization methods previously discussed in our earlier comment. All of the reported results for GABO do not include a filtering with proxy step. We apologize for the earlier confusion and believe the top-128 results above are exactly what the Reviewer is asking for.

---

> > > > > > > ### Author Response · Authors · 2024-08-12
> > > > > > >
> > > > > > > **On the inclusion of the Warfarin Task as a new offline MBO task:** We agree with the Reviewer that determining and proposing other conditional MBO tasks are important - because our primary contribution is algorithmic in nature (i.e., proposing SCR and GABO), we proposed one such conditional MBO to evaluate our contribution on, in additional to standard, "non-conditional" offline MBO tasks. We are happy to better highlight the nature of the Warfarin task (namely, that it features a convex oracle objective function and so first-order optimization methods work well out-of-the-box) in the discussion of the different tasks in our manuscript.
> > > > > > >
> > > > > > > We agree with the Reviewer that identifying new conditional MBO tasks is an important problem. As these new tasks are identified and proposed, SCR can be applied out-of-the-box to these tasks. In this work, we feel we have provided significant evidence already that our primary algorithmic contributions (i.e., GABO and SCR) are effective offline MBO tools that would be useful for the broader scientific community.
> > > > > > >
> > > > > > > **On including ROMO as a baseline method:** We appreciate the Reviewer sharing the ROMO work with us, and will be sure to include the comparison in the final manuscript.

---

> > > > > > > > ### Comment · Reviewer_yPeb · 2024-08-12
> > > > > > > >
> > > > > > > > Thank you for your thorough effort in active discussions. I hope that several discussion points will be reflected in the manuscript. I raised my score to 6, Best.

---

> > > > > > > > > ### Author Response · Authors · 2024-08-12
> > > > > > > > >
> > > > > > > > > Thank you for your helpful comments and feedback! We appreciate your support and thoughtful discussion.

---

### Official Review · Reviewer_AH6U · 2024-07-10

**Soundness:** 2
**Presentation:** 1
**Contribution:** 2
**Rating:** 5
**Confidence:** 4

**Summary:**

This work tackles offline Bayesian optimization using adaptive source critic regularization. The authors propose generative adversarial Bayesian optimization, which optimizes against a learned surrogate model without querying the true oracle function during optimization. It utilizes a Lipschitz-bounded source critic model to constrain the optimization trajectory. Finally the authors provide experimental results against various baseline methods.

**Strengths:**

- The setup of offline Bayesian optimization seems interesting.

**Weaknesses:**

- The rationale behind the proposed method is unclear.
- Details of experiments are missing.

**Questions:**

- Could you explain how these examples "Evaluating newly proposed molecules requires expensive experimental laboratory setups, and testing multiple drug doses for a single patient can potentially be dangerous" agree with your problem formulation?
- This sentence "Leveraging source critic model feedback for adversarial training of neural networks was first introduced by Goodfellow et al. (2014)" is not true. The authors should revise it.
- I think that this work just proposes a better regression method in the offline setting.
- Can you provide the details of $f_\theta$? Why did you choose such parameterization?
- How does a learned surrogate model provide additional information without querying the true oracle function? Source critic regularization doesn't add anything.
- When do you evaluate a true objective function? Didn't you evaluate query points when comparing your method to baseline methods?
- Equation (10): Is argmin correct? Should it be min?
- Can you explain the rationale behind $\lambda = \alpha / (1 - \alpha)$?
- Tables 1 and 2: How can the baseline methods show the results over the best in $\mathcal{D}$? I think that it is bounded in $\mathcal{D}$.
- Tables 1 and 2: What is the purpose of the row of the best? How can I interpret the results better than the best?
- It is a minor thing, but "bolded" is not a correct word. It should be just "bold."

**Limitations:**

There is no particular societal limitations of this work.

---

> ### Author Rebuttal · Authors · 2024-08-07
>
> We respond to Reviewer AH6U's comments below:
>
> **Rationale of SCR**
>
> To summarize our work and as discussed in our manuscript, **the primary rationale behind our proposed source critic regularization (SCR) method is to stop optimization algorithms from extrapolating against offline surrogate models in generative design.** We accomplish this by showing how to optimally and tractably balance the tradeoff between exploring the design space for potentially better designs while ensuring we don't over-extrapolate against the learned surrogate model.
>
> **Experimental Details**
>
> We have described the details of our experiments in **Section 5** and **Appendix A** of our manuscript, and also have made our code available for easy reproducibility and transparency. If the Reviewer has any remaining questions on experimental details, please feel free to let us know.
>
> **SCR is not a regression method.**
>
> To clarify, our work is not proposing a regression method; instead, **the primary contribution of SCR is to show how to balance the tradeoff between (1) optimization against a regressor model and (2) staying in-distribution in a computationally tractable way in offline MBO**.
>
> To validate this primary contribution empirically, we make use of generic, task-agnostic surrogate regression models trained using standard machine learning techniques (see **Section 5.2**, L271-5 for details). We do not optimize the hyperparameters of our learned surrogate regression models in any way, and use the same generic parametrization of $f_{\theta}$ across all tasks (L271-2). There are no special training methods that were used in constructing the surrogate regressor models. We are happy to clarify further if the Reviewer has any additional questions regarding the primary motivation and contributions of our work.
>
> > ***How does a learned surrogate model provide additional information without querying the true oracle function? Source critic regularization doesn't add anything.***
>
> The learned surrogate model provides information on the true oracle function because it is trained on historical designs and their corresponding true oracle function scores (represented as the dataset $\mathcal{D}$ in our manuscript). In regions of the design space that contain many examples of these historical designs in $\mathcal{D}$, the surrogate model agrees with the true oracle function because it has been trained on past observations of the true oracle function. However, in regions of the design space that do *not* contain many examples of historical designs, the surrogate model and true oracle function likely disagree due to the problem of **extrapolation**. A good illustration of this is shown in **Figure 1** of our manuscript.
>
> **The value added by source critic regularization is in preventing optimization algorithms from taking advantage of these extrapolated regions of the design space that result in falsely "optimal" designs.** In offline optimization, we want to avoid such designs that "look promising" according to the surrogate but in reality score poorly on the true oracle due to extrapolation of the surrogate model.
>
> > ***When do you evaluate a true objective function?***
>
> For each of the optimization methods (including GABO), we query the true objective function once (for **Table 1**) as the final step after an offline optimization algorithm is finished running and proposes a single design to evaluate using the hidden oracle objective function. This evaluation strategy is aligned with the motivation for this work; after we perform offline optimization experiments to propose a candidate design(s), we need to evaluate these designs with the actual oracle function to see how good these designs actually are. This evaluation schema is consistent with that used in most (if not all) other related work in offline generative design.
>
> **Reparametrization of $\lambda**
>
> The rationale behind re-parameterizing $\lambda$ in terms of $\alpha$ is that the feasible search space for $\lambda$ is all non-negative real numbers, which cannot be tractably searched over in a clear way. Our approach to solve this problem is to instead re-parameterize $\lambda$ in terms of $\alpha$ so that searching over a finite $\alpha\in[0, 1]$ is "equivalent" to searching over $\lambda\in[0, \infty)$.
>
> **Interpretation of $\mathcal{D}$(best) in Tables 1 and 2**
>
> Recall that $\mathcal{D}$ refers to the dataset of  historical designs and their corresponding true oracle function scores that was used to train the learn surrogate model. In Tables 1 and 2, $\mathcal{D}(\text{best})$ refers to the best oracle function score observed from this set of prior historical designs. For example, in **Table 1**, the value of $\mathcal{D}(\text{best})=11.3$ for the LogP task means that out of all the 79,564 unique molecules and their corresponding oracle LogP values in the base offline dataset associated with the LogP task, the maximum score achieved by any given molecule is 11.3.
>
> Therefore, if an optimization algorithm like GABO proposes a design with an oracle score greater than the best in $\mathcal{D}$ (e.g., a molcule with an oracle LogP score greater than 11.3), this means that we have "discovered" a new design not previously seen in the historical dataset that is even better than the best design in the historical dataset. Finding designs better than $\mathcal{D}(\text{best})$ is the main goal of offline optimization for generative design.
>
> Our choice of notation and in reporting $\mathcal{D}(\text{best})$ is consistent with that used in other related work ([Krishnamoorthy et al. Proc ICML 2023](https://arxiv.org/abs/2206.10786), [Krishnamoorthy et al. Proc ICML 2023](https://arxiv.org/abs/2306.07180), [Trabucco et al. Proc ICML 2021](https://arxiv.org/abs/2107.06882), [Chen et al. Proc NeurIPS 2022](https://arxiv.org/abs/2209.07507)).

---

> > ### Author Response · Authors · 2024-08-12
> >
> > Hi, hope you're doing well! Thank you again for your feedback and consideration of our manuscript. We hope that we have been able to answer your questions, and highlight how GABO and SCR may be a useful resource for the scientific community. We wanted to check if there are any remaining questions we can help address during this discussion period?

---

> > > ### Comment · Reviewer_AH6U · 2024-08-12
> > >
> > > Thank you for your response. I would update my score to 5. Please update your manuscript carefully considering my review and other reviews.

---

> > > > ### Author Response · Authors · 2024-08-12
> > > >
> > > > Thank you for your feedback and support of our work! We will be sure to update the manuscript based on the thoughtful comments from all reviewers.

---

### Official Review · Reviewer_Yd2B · 2024-07-12

**Soundness:** 3
**Presentation:** 2
**Contribution:** 3
**Rating:** 6
**Confidence:** 4

**Summary:**

This paper considered an offline optimization problem where a surrogate objective function instead of the oracle objective can be queried. The surrogate objective function is trained using a reference offline dataset and thus may falsely predict the optimum due to overestimation errors. To resolve this issue, this paper proposed a Generative Adversarial Bayesian Optimization (GABO) algorithm which exploits the adaptive Source Critic Regularization (SCR) to achieve robust oracle function optimization. The proposed method is demonstrated to outperform other tested offline optimization baselines on several synthetic and real-world datasets.

**Strengths:**

1. The idea of adding a regularization penalty to the original offline optimization problem is interesting. The technical issues are clearly introduced.

2. The empirical performance improvement of the proposed GABO is significant compared to the other BO and model-based optimization algorithms.

**Weaknesses:**

I have reviewed this paper before and had multiple rounds of discussions with the authors in the previous review process. My major concern is still about the motivation and positioning of this work. The key contribution of this work is the design of the surrogate objective in (7) and its dual formulation in (8) for a model-based optimization (MBO) problem. This new objective is independent of BO and can actually be optimized using the zero-order, one-order, or any other optimization algorithm. Even though this paper has highlighted and empirically shown that BO is an effective choice among multiple optimization methods, I still think the combination of SCR and BO is trivial and should not be the focus of this work.

I cannot consider the proposed GABO algorithm as a novel **BO** framework since all the new techniques proposed are not specifically designed for BO. The challenges of applying BO in the generative adversarial optimization problem are unclear. I highly suggest the authors to reconsider the title and motivation of this work such that its contributions can be clearer.

**Questions:**

In Lines 210-213, it is mentioned that the objective of GABO is time-varying, which is an interesting and non-trivial issue of applying BO to this problem. Do you have any idea about how to resolve this issue?

**Limitations:**

This limitations are clearly discussed.

---

> ### Author Rebuttal · Authors · 2024-08-07
>
> We thank Reviewer Yd2B for their thoughtful review of our work, and appreciate that they recognize our proposed source critic regularization (SCR) formulation and strategy as a key contribution of our work.
>
> **Framing of the Manuscript's Narrative**
>
> We agree that our main contribution is Source Critic Regularization (SCR) in **Algorithm 1**, which is optimizer-agnostic and can be used with any optimization method. Our experimental evaluation of SCR using Bayesian Optimization (BO) is because we find it to be the most effective optimizer to use in conjunction with SCR, likely due to BO's well-documented effectiveness as an offline optimizer as outlined in **Section 3.2** (L110-7). Indeed, as acknowledged by the Reviewer, we have already included evidence of the effectiveness of BO compared to alternatives such as Gradient Ascent (GA)$\textemdash$in particular, our comparison to Generative Adversarial Gradient Ascent (GAGA) in **Supplementary Table B5**. In addition, GABO also outperforms vanilla BO in all our experiments, demonstrating that SCR has significant added value as a methodological contribution.
>
> In alignment with the feedback from the Reviewer, we are happy to revise the title of our work to the following: **Generative Adversarial Model-Based Optimization via Source Critic Regularization**. This improvement will help better emphasize that our main contribution is SCR independent of the choice of baseline optimizer. We will also do our best to revise our paper to make it more clear that SCR is our main contribution, and that BO is simply the most effective instantiation of our method.
>
> **Future Work on Time-Varying Optimization**
>
> Thank you for this comment; indeed time-varying BO is an interesting and nascent field of research and a complex problem in general. We hypothesize that there may be opportunities to exploit existing methods in time-varying BO for SCR if the updates to the source critic function $c^*$ follow some sort of pattern over time. In practice, we were unable to identify any such patterns in preliminary studies largely due to the limited number of updates to $c^*$ made over the course of the optimization process. Furthermore, factoring in time-varying BO was not necessary to demonstrate that GABO and SCR are effective empirically. Given the complexity of this problem and the empirical success of GABO and SCR already demonstrated in our work, we leave this opportunity to explore time-varying BO for future work.

---

> > ### Author Response · Authors · 2024-08-12
> >
> > Hi, hope you're doing well! Thank you again for your continued consideration of our manuscript. We hope that the title change and focus of SCR as the primary contribution of our work better align with your helpful feedback. We have also included [new results](https://openreview.net/forum?id=3RxcarQFRn&noteId=n2sQJLH7uv) comparing both GABO and GAGA against other baseline optimization methods to help readers better assess the utility of SCR in the main text of our updated manuscript. We wanted to check if there are any remaining questions we can help address during this discussion period?

---

> > > ### Comment · Reviewer_Yd2B · 2024-08-12
> > >
> > > The author responses have addressed my major concern. Therefore, I raised my score as 6.

---

> > > > ### Author Response · Authors · 2024-08-12
> > > >
> > > > We appreciate your helpful comments and support of our work. Thank you!

---

### Official Review · Reviewer_bMXC · 2024-07-15

**Soundness:** 3
**Presentation:** 3
**Contribution:** 3
**Rating:** 7
**Confidence:** 4

**Summary:**

The paper considers the problem of offline black-box optimization where only limited (zero-shot or few-shot) online interactions with the objective function is available. Existing approaches commonly train a neural net parametrized surrogate model of the objective using the offline data. The paper proposes to use a source critic model, inspired by discriminator training in generative adversarial networks, to regularize this surrogate. This is accomplished by formulating a constrained optimization problem which constrains the optimization (over the surrogate model) trajectory samples to be similar to the training data using the source critic model. The Lagrangian version of this constrained optimization is used as the objective for a standard batch mode Bayesian optimization algorithm. Experiments are performed on tasks from Design-bench benchmark and a new Warfarin task.

**Strengths:**

- One thing I really like about the paper is its evaluation which truly mirrors the offline optimization setting. The paper compares the proposed approach and baselines on a single evaluation from the oracle which I believe is the right way to evaluate algorithms for offline model-based optimization.

- The proposed approach shows good performance across all the tasks on both zero-shot and few-shot evaluation. The ablation study in 5.5 also shows that adaptively tuning the lagrangian coefficient is also important for good performance.

- The constrained optimization formulation is well-formulated and principled approach to tackle this problem. The design choice of constraining the surrogate model using the source critic model makes sense and is relevant for the problem.

- Some of the tasks in design-bench benchmark have multiple errors which makes them not so informative for evaluation. I like the paper didn't include superconductor task where the original offline dataset itself has multiple copies of the same inputs but with different outputs. Similarly, I think the ChemBL and UTR tasks are also not useful. In my practical experience, samples searched over ChemBL generates a lot of syntactic errors. I like the fact that the paper introduces new tasks like LogP and Warfarin which will be useful to the broader community.

**Weaknesses:**

- One question I have is about the broader picture of using Bayesian optimization (BO) for this problem space. Why should we first train the neural network surrogate model and then fit the Gaussian process (GP) on top of the neural surrogate model to do few steps of BO? Why can't we just fit a GP (with a latent space presumably to handle high/structured dimensionality) directly on the offline data and do one step of BO? I am assuming the TurBO-qEI baseline doesn't do that and also fits the Turbo's GP on the neural network surrogate.

- It would be nice to see more justification for the BO hyperparameters (like sampling budget (T) and batch size (b)). Is the method sensitive to the choice of these parameters?

- One thing I like about the paper is that it can work with any choice of the surrogate model. I would also like to point out one very recent related work that also constrains the optimization given any surrogate model by formulating the search as an offline RL problem.

        - Yassine Chemingui, Aryan Deshwal, Nghia Hoang, Janardhan Rao Doppa. Offline Model-based Black-Box Optimization via Policy-guided Gradient Search. Thirty-Eighth AAAI Conference on Artificial Intelligence (AAAI), 2024.

**Questions:**

Please see weaknesses section.

**Limitations:**

Please see weaknesses section.

---

> ### Author Rebuttal · Authors · 2024-08-07
>
> We thank Reviewer bMXC for their thoughtful comments and insights, and share their enthusiasm for the proposed significance and strengths of our work. We address their outstanding questions in our response below.
>
> **The Role of the Surrogate NN in MBO**
>
> We thank the reviewer for this comment. We use this setup for generality; in many applications, the surrogate objective is not under our control. For instance, it might be a physics simulator or other domain-specific function that has been optimized by domain experts to give more information about the hidden oracle function across the entire design space than what can be accomplished with GPs alone. (Note that task-specific optimization of the surrogate is *not* done in our work.) By fitting a GP to this surrogate instead of just the observations of the hidden oracle function, we can take advantage of the information encoded in a more complex (and hopefully accurate) surrogate.
>
> To demonstrate this empirically, we explore this potential strategy of using the single GP from BO directly as the both the surrogate function *and* for BO sampling. We leverage Source Critic Regularization (i.e., **Algorithm 1**) using this framework. We refer to this as Generative Adversarial Bayesian Optimization with a GP Surrogate (GABO GP-Surrogate), and compare this method with GABO:
>
> | One-Shot Oracle Evaluation | Branin          | LogP            | TFBind            | GFP             | UTR             | ChEMBL          | Warfarin | Avg. Rank |
> | -------------------------- | ------ | ---- | ------ | --- | --- | ------ | -------- | --- |
> | **GABO GP-Surrogate**          | -37.4 $\pm$ 4.4 | -57.9 $\pm$ 159 | **0.576 $\pm$ 0.058** | 3.51 $\pm$ 0.69 | 6.84 $\pm$ 1.24 | **0.65 $\pm$ 0.01** | -0.27 $\pm$ 2.13 | 7.7 |
> | **GABO**                       | **-2.6 $\pm$ 1.1** | **21.3 $\pm$ 33.2** | 0.570 $\pm$ 0.131 | **3.60 $\pm$ 0.40** | **7.51 $\pm$ 0.39** | 0.60 $\pm$ 0.07 | **0.60 $\pm$ 1.80** | **4.3** |
>
> | $k=64$-Shot Oracle Evaluation | Branin          | LogP            | TFBind            | GFP                | UTR             | ChEMBL          | Warfarin | Rank |
> | ----------------------------- | ------ | ---- | ------ | --- | --- | ------ | -------- | --- |
> | **GABO GP-Surrogate**             | -8.4 $\pm$ 1.6  | 0.66 $\pm$ 0.01 | 0.720 $\pm$ 0.068 | 3.74 $\pm$ 0.00 | **8.27 $\pm$ 0.08** | 0.66 $\pm$ 0.01 | 0.98 $\pm$ 0.05 | 6.6 |
> | **GABO**                          | **-0.5 $\pm$ 0.1** | **98.0 $\pm$ 37.6** | **0.942 $\pm$ 0.026** | **3.74 $\pm$ 0.00** | 8.25 $\pm$ 0.17 | **0.67 $\pm$ 0.02** | **1.00 $\pm$ 0.28** | **2.1** |
>
> As we can see, using a more complex, neural-network surrogate function for GABO leads to better optimization results than directly using the GP as the surrogate function.
>
> Our evaluation strategy of using an neural network surrogate for GABO is also identical to the one used in the existing offline MBO literature to make it easier to compare our work with pre-existing methods: for instance, [Trabucco et al. (ICML 2021)](https://arxiv.org/abs/2107.06882), [Yu et al. (NeurIPS 2021)](https://arxiv.org/abs/2110.14188), [Chen et al. (ICML 2023)](https://arxiv.org/abs/2301.02931), [Krishnamoorthy et al. (ICML 2023)](https://arxiv.org/abs/2306.07180), and [Krishnamoorthy et al. (ICML 2023)](https://arxiv.org/abs/2206.10786) in addition to our work.
>
> **Sensitivity to Sampling Budget and Batch Size**
>
> Thank you for this comment. Prior work has shown that BO methods are relatively robust to perturbations in the batch size and other hyperparameters (i.e., Figure 7 in [Eriksson et al. (NeurIPS 2019)](https://arxiv.org/abs/1910.01739)), and we have observed that this similarly applies to GABO. We will include a discussion on the sensitivity to the optimization hyperparameters in an updated version of our manuscript.
>
> **Additional Citation**
>
> Thank you for sharing this related work from Chemingui et al. with us. We will include a citation to this work and appropriate associated discussion in the final manuscript.

---

> > ### Comment · Reviewer_bMXC · 2024-08-11
> > **Response to rebuttal**
> >
> > Thanks for the response to my questions. I am happy with the response and believe this paper will be a good addition to the offline model based optimization literature. Hence, I will keep my positive score.

---

> > > ### Author Response · Authors · 2024-08-11
> > >
> > > Thank you for your careful consideration and support of our work!

---

### Official Review · Reviewer_U2YG · 2024-07-16

**Soundness:** 4
**Presentation:** 4
**Contribution:** 3
**Rating:** 6
**Confidence:** 5

**Summary:**

This paper proposes a novel approach for offline MBO that combines Source Critic Regularization and Bayesian Optimization. Offline MBO aims to train a surrogate model from offline data and subsequently extrapolates it to find an optimal design (as opposed to online MBO which collects more data as it trains the surrogate model). One key challenge of offline MBO is knowing when to trust the surrogate model, since it could be highly erroneous at ood inputs. This paper handles said challenge by incorporating a critic constraint that penalizes candidate that are not sufficiently similar to the training samples.

**Strengths:**

- Clear description, easy to follow
- Practical and novel method for MBO. As a regularization method it can be incorporated into many other MBO frameworks.
- Hyperparameter \lambda is chosen methodically
- Perform well on 100th percentile metric

**Weaknesses:**

- Doesn't seem to be very good on 90th percentile metric (i.e., mean rank ~ 8). Does this mean that the method is very good at picking out high-reward outlier, but the overall quality of the surrogate is about the same as everything else? If that's the case is there any justification for this behavior? I would prefer if the authors plotted out the entire scatter plot of 100 candidates for each method, as that would allow us to compare performance on a distributional level.

**Questions:**

- Why run BO on the learned surrogate when it is already differentiable and can be numerically optimized using GA? (I saw the ablation study on GAGA but just want to hear an argument for the benefit of GABO).

**Limitations:**

No potential negative societal impact

---

> ### Author Rebuttal · Authors · 2024-08-07
>
> We thank Reviewer U2YG for their thoughtful feedback on our work, and address their outstanding questions and concerns below.
>
> **Performance of GABO on the 90th percentile metric**
>
> Unlike the other evaluation metrics assessed in our work, GABO indeed does not perform as well on the 90th percentile metric. However, this is not surprising since GABO is not designed to target this metric (and it is not our metric of interest). In particular, in our analysis, we have found that this is largely due to the nature of the underlying Bayesian optimization (BO) optimization algorithm. Because BO is not an iterative first-order algorithm, the designs proposed by any BO-based algorithm often have high variance in practice. This is what we observe across all of our experiments, including in **Table 1** and **Supplementary Tables B1 and B3**.
>
> We also note that in most offline optimization applications, the 90th percentile metric$\textemdash$or any metric that does not use for the best proposed design(s)$\textemdash$is not as useful as the other metrics assessed where GABO *does* perform well. This is because in offline optimization tasks with a restricted budget to query the hidden, expensive-to-evaluate oracle function, we are not interested in "wasting" this budget on subpar design candidates. While the 90th percentile and similar metrics can be helpful to understand the limitations of algorithms (as in this case), we believe that the 100th percentile metrics reported in the main text are more useful and practical in assessing each of the optimization algorithms.
>
> For completeness, we also show the distributions of the proposed designs for across all of the optimization methods for the LogP task to allow interested readers to compare the performance of the different methods on a distributional level. The plot is available here: [Link](https://bashify.io/files/dxnMON). We will include this result in the final manuscript.
>
> **Why run BO on the learned surrogate when it is already differentiable and can be numerically optimized using GA?**
>
> Thank you for this question. We choose to use BO for the basis of our work because it is a well established principle in recent Bayesian Optimization (BO) literature that BO is useful even for differentiable objectives, outperforming baselines such as GA [Eriksson et al. (NeurIPS 2019)](https://arxiv.org/abs/1910.01739), [Maus et al. (NeurIPS 2022)](https://arxiv.org/abs/2201.11872), [Hvarfner et al. (2024)](https://arxiv.org/abs/2402.02229), [Eriksson et al. (UAI 2021)](https://proceedings.mlr.press/v161/eriksson21a/eriksson21a.pdf), [Astudillo et al. (ICML 2019)](https://arxiv.org/abs/1906.01537). It is the dimensionality and non-convexity of the search space that makes the optimization problems in our benchmark challenging, leading first order methods like GA to struggle. Based on these results, we believe BO is a natural basis for our work. We discuss this in detail in **Section 3.2** (L110-7).
>
> Nevertheless, we emphasize that our proposed source critic regularization (SCR) algorithm can be used with other optimization methods as well, such as Gradient Ascent (GA) as suggested. Indeed, as already noted by the Reviewer, we also evaluated SCR with GA, which we call Generative Adversarial Gradient Ascent (GAGA), and include results in **Supplementary Table B5**. Comparing GABO with GAGA helps motivate our decision to use BO for experimental evaluation of our SCR algorithm. To better emphasize that our main contribution is SCR, we plan to revise the title of our work to the following: **Generative Adversarial Model-Based Optimization via Source Critic Regularization**

---

> > ### Author Response · Authors · 2024-08-12
> >
> > Hi, hope you're doing well! Thank you again for your feedback and consideration of our manuscript. We hope that (1) the additional distribution plot results included in the summary of revisions; and (2) the detailed rationale of why we evaluate SCR with BO has helped address your initial concerns. We wanted to check if there are any remaining questions we can help address during this discussion period?

---

> > > ### Comment · Area_Chair_TdqN · 2024-08-13
> > >
> > > Dear reviewer, please read the above rebuttal and evaluate whether it answers your concerns. If your evaluation remains unchanged, please at least acknowledge that you have read the author's response.

---

> > > ### Comment · Reviewer_U2YG · 2024-08-13
> > > **Thanks for the response**
> > >
> > > And sorry for the late chiming in.
> > >
> > > Regarding your point that the 90 percentile metric is not as useful as the 100 percentile. While I generally agree with this opinion, it is only true if you have a budget of 100 trials to find your best solution. If you have exactly one trial, the 90 percentile, or even 50 percentile metric helps to show that there is low risk of your one shot being a bust (because without evaluation how do you know which one among the 100 is the best?). On the other hand, if you do have 100 trials, why not simply cast this as an online optimization problem?
> > >
> > > I feel like this is a minor point (which I am more than happy to continue discussing), and will keep my score regardless.

---

> > > > ### Author Response · Authors · 2024-08-14
> > > >
> > > > Thank you for this question and additional opportunity for discussion! Because GABO leverages the penalized surrogate model to rank candidate designs, the candidate is that the single 100th percentile-ranked candidate typically outperforms the single 90th percentile-ranked candidate according to the true oracle, as is demonstrated by our experiments. We also believe that another contributing factor is that BO-based optimization methods generally struggle according to this evaluation metric due to their high variance, as evidenced by our reported BO-qEI and TuRBO-qEI results. Finally, the new top-128 evaluation results we have performed - and GABO's strong performance on this metric - also help support its utility in generative design. Regardless, we appreciate the Reviewer's continued support of our manuscript and thoughtful feedback throughout this review process.

---

### Official Review · Reviewer_czFU · 2024-07-29

**Soundness:** 3
**Presentation:** 3
**Contribution:** 2
**Rating:** 6
**Confidence:** 4

**Summary:**

This paper proposes to use an adversarial regulariser in the BO setting. In particular, the authors propose a systematic way to compute the regularization parameter through a lagrange duality. Overall the regularized method performs on average better than existing methods across a suite of benchmark datasets.

**Strengths:**

- The paper is the first to propose an adversarial regularization objective for BO.
- The method is well motivated and a practical way to obtain the hyperparameter is given, albeit through a grid.
- The proposed methods performs on par with baselines and on average is better across several benchmark datasets.

**Weaknesses:**

- to clarify, the way you choose alpha is through a grid. Wouldnt that mean your computational cost is significantly larger than baseline methods, please add a section on computational time as well as cost to better understand the tradeoffs of the proposed method over existing methods. Please comment on the above and clarify any misunderstandings please. Especially this part: "To ensure a fair evaluation schema, all MBO methods were evaluated using a fixed surrogate query budget of 2048". Can you confirm that no extra data was used to obtain the alpha in your algorithm? There might be a misunderstanding here, in particular what how exactly to pick the best alpha based on the grid of 200 and the associated costs?
- VAEs as well as GAN are often time hard to train and need very specific architectures to work well. How, were these chosen and is there any ablation study? how does the quality of "c" affect the problem setting? My concern is that there are so many moving parts that is becomes hard to understand what the contribution of each part is
- I dont understand D(best) how come some values in the table are able to be better than the oracle? Am i reading the table correctly?


I am more than happy to change my score if the above have been clarified

**Questions:**

see above

**Limitations:**

yes

---

> ### Author Rebuttal · Authors · 2024-08-07
>
> We thank Reviewer czFU for their thoughtful review of our work, and appreciate that they recognize the technical and empirical contributions of GABO compared with prior work. Please find our responses to their questions below.
>
> **Search for $\alpha$**
>
> Importantly, we confirm that no extra data is used to compute the value of $\alpha$ in our algorithm -- because only the source critic $c^*$ is retrained dynamically over the optimization process, we only need to query the easy-to-compute neural network $c^*$ in computing $\alpha$.
>
> We also confirm that we compute $\alpha$ through a grid search. However, in our implementation used for our experiments (which will be made publicly available), the grid search to compute $\alpha$ is highly vectorized such that the computational time is non-limiting even using a single-GPU setup. To benchmark our implementation, we evaluate both BO and Gradient Ascent (GA) both with and without our Generative Adversarial (GA) source critic regularization algorithm on the Branin and Penalized LogP optimization tasks:
>
> | **Method** | **Branin Task Compute Time (sec)** | **LogP Task Compute Time (sec)** |
> | --- | --- | --- |
> | BO-qEI | 92.1 $\pm$ 10.2 | 965 $\pm$ 17.9 |
> | GABO | 328 $\pm$ 146 | 1245 $\pm$ 55.2 |
> | GA | 9.68 $\pm$ 0.23 | 765 $\pm$ 6.64 |
> | GAGA | 75.6 $\pm$ 25.4 | 818 $\pm$ 10.5 |
>
> Compute time is reported as mean $\pm$ standard deviation over 10 random seeds. As a reminder, the Branin task is a standard benchmarking task for offline optimization, and the Penalized LogP task is subjectively the most challenging task assessed in our manuscript with the highest dimensional design space out of the seven assessed tasks.
>
> While there are obviously additional compute costs associated with running our Source Critic Regularization method (i.e., **Algorithm 1**), as made evident by the above results, we note that in most applications of offline optimization, obtaining labeled data is the main bottleneck in many practical applications; thus, it is often worth spending this extra compute to ensure the best results for the given budget.
>
> **Selection of Model Hyperparameters**
>
> The exact same generic FCNN-based source critic as introduced by the initial WGAN authors [Arjovsky et al. (2017)](https://arxiv.org/abs/1701.07875) is used across all of the different tasks assessed in our work. Similarly, we perform **no** hyperparameter fine-tuning of VAEs and use standard task-agnostic VAE architectures across the different tasks in our work. By using generic model architectures that have *not* been optimally tuned to each task, we can focus on assessing our contributions that are primarily algorithmic in nature. That is, the VAEs and source critic models may perform well on certain tasks and may perform poorly on others that were assessed. By evaluating our source critic regularization algorithm across a wide variety of different benchmarking tasks, we are able to give a good picture of the "average" performance of GABO and other optimizers independent of how well VAEs and source critics are tuned to the specific optimization task at hand.
>
> **What is D(best)?**
>
> In our results, $\mathcal{D}$(best) refers to the best oracle score achieved by any previously observed design in the offline dataset. For example, in **Table 1**, the value of $\mathcal{D}_{\text{best}}=11.3$ for the LogP task means that out of all the 79,564 unique molecules and their corresponding oracle LogP values in the base offline dataset associated with the LogP task, the maximum score achieved by any given molecule is 11.3. Therefore, if an optimization method (such as GABO) proposes a molecule design that achieves an oracle LogP score greater than 11.3, then we have found a design better than any of the designs previously seen in the offline dataset. Our choice of notation is consistent with that used in related work ([Krishnamoorthy et al. Proc ICML 2023](https://arxiv.org/abs/2206.10786), [Krishnamoorthy et al. Proc ICML 2023](https://arxiv.org/abs/2306.07180), [Trabucco et al. Proc ICML 2021](https://arxiv.org/abs/2107.06882), [Chen et al. Proc NeurIPS 2022](https://arxiv.org/abs/2209.07507)).

---

> > ### Comment · Reviewer_czFU · 2024-08-08
> > **response**
> >
> > Thanks for the rebuttal.
> > All my questions have been addressed and hence i am happy to increase my score to 6.

---

> > > ### Author Response · Authors · 2024-08-08
> > >
> > > We are grateful for Reviewer czFU's thoughtful consideration and support of our manuscript. Thank you!

---

### Author Rebuttal · Authors · 2024-08-07

# Summary of Revisions Made to the Paper

We thank the Reviewers for their thoughtful comments and consideration of our paper. We are grateful that the Reviewers find our method novel (Reviewer czFU, bMXC, Yd2B, yPeB), well-justified (Reviewer bMXC, Yd2B, yPeB), well-written (Reviewer U2YG), and a useful resource (Reviewer bMXC, yPeB) for the broader scientific community.

We have the following general comments, including several changes we have made to our manuscript to address reviewer comments:
  1. Several reviewers asked about the importance of Bayesian Optimization (BO) in our framework. Indeed, our main contribution is to use Source Critic Regularization (SCR) for Model Based Optimization (MBO), as illustrated in **Algorithm 1**. While SCR can be applied to several different optimization methods, we find BO to be the most effective (**Section 3.2**, **Supplementary Table B5**). This motivates using BO as our vehicle to evaluate SCR. To illustrate that SCR can also be used with other optimization methods, we have also demonstrated how SCR can be used to improve upon Gradient Ascent, resulting in Generative Adversarial Gradient Ascent (GAGA). To better emphasize that our main contribution is SCR, we are happy to change our title to **Generative Adversarial Model-Based Optimization via Source Critic Regularization**.
  2. We have included a plot of the distribution of scores for the Penalized LogP task for different optimizers in the attached PDF file for the interested reader to analyze how the distribution of oracle scores vary across different methods.
  3. We have included further justification for our top-1 evaluation of designs using the oracle, which is different from prior work as noted by some Reviewers. Similar to Reviewer bMXC, we believe that this evaluation metric is "right way to evaluate algorithms for offline model-based optimization." Nevertheless, we emphasize that we have already included top-64 scores in **Supplementary Tables B1 and B2**, which is a "more common" metric in related work. In fact, GABO performs even better in terms of the top-64 metric than it does at our top-1 metric; our focus on top-1 is solely because we believe it to be more relevant to real-world tasks.
  4. We have included additional experimental details regarding the computational time and resources to run our experiments compared with baseline optimization methods, which we discuss in more detail in responses to individual reviewers.

We look forward to discussing further with the Reviewers to answer any outstanding questions or concerns. Thank you!

---

### Comment · Area_Chair_TdqN · 2024-08-08

Dear authors and reviewers,

The authors-reviewers discussion period has now started.

@Reviewers: Please read the authors' response, ask any further questions you may have or at least acknowledge that you have read the response. Consider updating your review and your score when appropriate. Please try to limit borderline cases (scores 4 or 5) to a minimum. Ponder whether the community would benefit from the paper being published, in which case you should lean towards accepting it. If you believe the paper is not ready in its current form or won't be ready after the minor revisions proposed by the authors, then lean towards rejection.

@Authors: Please keep your answers as clear and concise as possible.

The AC

---

### Decision · Program_Chairs · 2024-09-25

**Decision:**

Accept (poster)

**Comment:**

The reviewers unanimously recommend acceptance (6-6-7-6-5-6). The author-reviewer discussion has been constructive and has led to a number of improvements to the paper, regarding the presentation, the framing of contribution, as well as the empirical evaluation. Proposed changes and new results have been discussed during the rebuttal. The authors are asked to now implement the changes discussed with the reviewers in the final version of the paper.